# Plural molecular and cellular mechanisms of pore domain *KCNQ2* encephalopathy

Timothy J Abreo[1,2], Emma C Thompson[1†], Anuraag Madabushi[1†], Kristen L Park[3,4†], Heun Soh[5], Nissi Varghese[5], Carlos G Vanoye[6], Kristen Springer[5], Jim Johnson[7], Scotty Sims[7], Zhigang Ji[1], Ana G Chavez[1,8], Miranda J Jankovic[1], Bereket Habte[3,4], Aamir R Zuberi[9], Cathleen M Lutz[9], Zhao Wang[10,11,12], Vaishnav Krishnan[1,8,13], Lisa Dudler[14], Stephanie Einsele-Scholz[14], Jeffrey L Noebels[1,2,8], Alfred L George[3], Atul Maheshwari[1,8], Anastasios Tzingounis[5], Edward C Cooper[1,2,8]*

[1]Department of Neurology, Baylor College of Medicine, Houston, United States; [2]Department of Molecular and Human Genetics, Baylor College of Medicine, Houston, United States; [3]Department of Neurology, Children's Colorado, University of Colorado, Aurora, United States; [4]Department of Pediatrics, Children's Colorado, University of Colorado, Aurora, United States; [5]Department of Physiology and Neurobiology, University of Connecticut, Storrs, United States; [6]Department of Pharmacology, Northwestern University Feinberg School of Medicine, Chicago, United States; [7]KCNQ2 Cure Alliance, Denver, United States; [8]Department of Neuroscience, Baylor College of Medicine, Houston, United States; [9]The Rare Disease Translational Center & Technology Evaluation and Development, The Jackson Laboratory, Bar Harbor, United States; [10]Department of Biochemistry and Molecular Pharmacology, Baylor College of Medicine, Houston, United States; [11]CryoEM Core, Baylor College of Medicine, Houston, United States; [12]Department of Molecular and Cellular Biology, Baylor College of Medicine, Houston, United States; [13]Department of Psychiatry and Behavioral Sciences, Baylor College of Medicine, Houston, United States; [14]Center for Human Genetics Tübingen, Tübingen, Germany

*For correspondence:
ecc1@bcm.edu

†These authors contributed equally to this work

## eLife assessment

The paper investigates a potential cause of a type of severe epilepsy that develops in early life because of a defect in a gene called KCNQ2. The significance is **fundamental** because it substantially advances our understanding of a major research question. The strength of the evidence is **convincing** because appropriate methods are used that are in line with the state-of-the art.

**Abstract** *KCNQ2* variants in children with neurodevelopmental impairment are difficult to assess due to their heterogeneity and unclear pathogenic mechanisms. We describe a child with neonatal-onset epilepsy, developmental impairment of intermediate severity, and *KCNQ2* G256W heterozygosity. Analyzing prior KCNQ2 channel cryoelectron microscopy models revealed G256 as a node of an arch-shaped non-covalent bond network linking S5, the pore turret, and the ion path. Co-expression with G256W dominantly suppressed conduction by wild-type subunits in heterologous cells. Ezogabine partly reversed this suppression. *Kcnq2*[G256W/+] mice have epilepsy leading to premature deaths. Hippocampal CA1 pyramidal cells from G256W/+ brain slices showed hyperexcitability. G256W/+ pyramidal cell KCNQ2 and KCNQ3 immunolabeling was significantly shifted from axon initial segments to neuronal somata. Despite normal mRNA levels, G256W/+ mouse KCNQ2 protein levels were reduced by about 50%. Our findings indicate that G256W pathogenicity results

from multiplicative effects, including reductions in intrinsic conduction, subcellular targeting, and protein stability. These studies provide evidence for an unexpected and novel role for the KCNQ2 pore turret and introduce a valid animal model of *KCNQ2* encephalopathy. Our results, spanning structure to behavior, may be broadly applicable because the majority of *KCNQ2* encephalopathy patients share variants near the selectivity filter.

## Introduction

*KCNQ2* variants are among the most frequent diagnostic findings from genetic tests for epilepsy in young children (*Symonds et al., 2019*; *Truty et al., 2019*). Such test results often raise new questions about pathogenicity and developmental prognosis. This uncertainty is partly due to the great diversity of different alleles known among individuals seeking care (n=1954 in NCBI ClinVar; *Landrum et al., 2018*). A second contributor is the broad phenotypic spectrum associated with *KCNQ2* variants (*Weckhuysen and George, 2022*). At the milder end, individuals may have seizures restricted to the first weeks or months of life and good later development—a disorder called self-limited familial neonatal epilepsy (SLFNE; *Ronen et al., 1993*; *Scheffer et al., 2017*). At the most highly impaired end, individuals have treatment-refractory, neonatal-onset seizures accompanied by lifelong profound global disability, a disorder first called '*KCNQ2* encephalopathy' (*Weckhuysen et al., 2012*), with electroclinical features akin to the older diagnostic group with very poor prognosis, Ohtahara syndrome (*Beal et al., 2012*; *Olson et al., 2017*). Case series reveal a middle group with de novo missense or small indel variants where seizures may remit early, and considerable childhood development of motor, receptive language, and social abilities takes place, albeit with delay. For many such affected people, spoken language remains absent, and other features including autism spectrum disorder, recurring if infrequent convulsions, and inability to perform activities of daily life independently impose significant limitations (*Weckhuysen et al., 2013*; *Millichap et al., 2016*). This has led to the concept of a *KCNQ2* developmental and epileptic encephalopathy (DEE) spectrum (*Dirkx et al., 2020*; *Berg et al., 2021*). Efforts to correlate developmental prognosis to variant functional consequences within critical structural domains have been made (*Millichap and Cooper, 2012*; *Millichap et al., 2016*; *Goto et al., 2019*; *Zhang et al., 2020*; *Brünger et al., 2023*), but prediction can likely be helped by more confluent biological evidence.

Voltage-gated K⁺ (Kv) channels contain voltage-sensing (VSD) and pore-gating (PGD) structural domains. During channel activity, conformational changes in the two domains are coupled giving rise to ion current. Here we analyze KCNQ2 G256W, a PGD variant found in an infant whose neonatal seizures remitted in early infancy and who subsequently gained neurodevelopmental milestones on an intermediate severity trajectory. G256W maps to the PGD turret near the top of the S5 helix. The G256 location is far from the primary channel components needed for ion flow (voltage sensor, pore gate, and selectivity filter), raising questions about pathogenicity and (if pathogenic) mechanisms. Prior biophysical studies of the K⁺ channel PGD turret have highlighted its roles in display of negative electrostatic surface charge, and its binding of pore-blocker venom toxins (*Miller et al., 1985*; *MacKinnon and Yellen, 1990*; *Hille et al., 1999*; *Banerjee et al., 2013*; *Zhao et al., 2019*). By examining and comparing recent models of KCNQ1, 2, and KCNQ4 generated by cryoelectron microscopy (cryoEM), we found evidence that the KCNQ2 G256 residue contributes to a previously unstudied KCNQ channel turret role, stabilizing the open selectivity filter from its extracellular side. We analyzed KCNQ2 G256W pathogenicity via a multilevel experimental program including expression in heterologous cells and Crispr/Cas9-generated knock-in mice. We also made mice with a neighboring frameshift variant, and directly compared this model of the milder SLFNE phenotype with G256W in vivo. Unlike many other Kv currents, KCNQ2 mediated M-currents are non-inactivating (*Brown and Adams, 1980*). Our results led us to conclude that the biological role of this distinctive $I_M$ property–absence of inactivation–is central to the pathophysiology of PGD variants like G256W.

## Results

### Clinical and developmental history of individual 1 index child

A non-dysmorphic female was born at term in the United States to parents of one older well-developing child. The neonate was vigorous with Apgar scores of 7 and 9, weight of 3253 gm (51.3%),

length of 48 cm (26.8%), and orbitofrontal head circumference of 34.5 cm, (69.7%). Epileptic seizures began 3 hr after birth, and included arching, facial flushing, head deviation, eye rolling, and upper extremity flexion with tensing. The mother reported events during the third trimester concerning for in utero seizures. The fetus would maintain a rigid posture for 20–30 min, and could not be repositioned, causing significant maternal pain. Each episode was followed by a period of decreased fetal movement. The newborn was treated with levetiracetam (20 mg/kg BID). As seizures persisted, the following day phenobarbital was added (2x10 mg/kg). Seizures continued, despite levetiracetam, phenobarbital, and subsequently, lorazepam. Brain magnetic resonance imaging and screening metabolic labs showed no abnormalities.

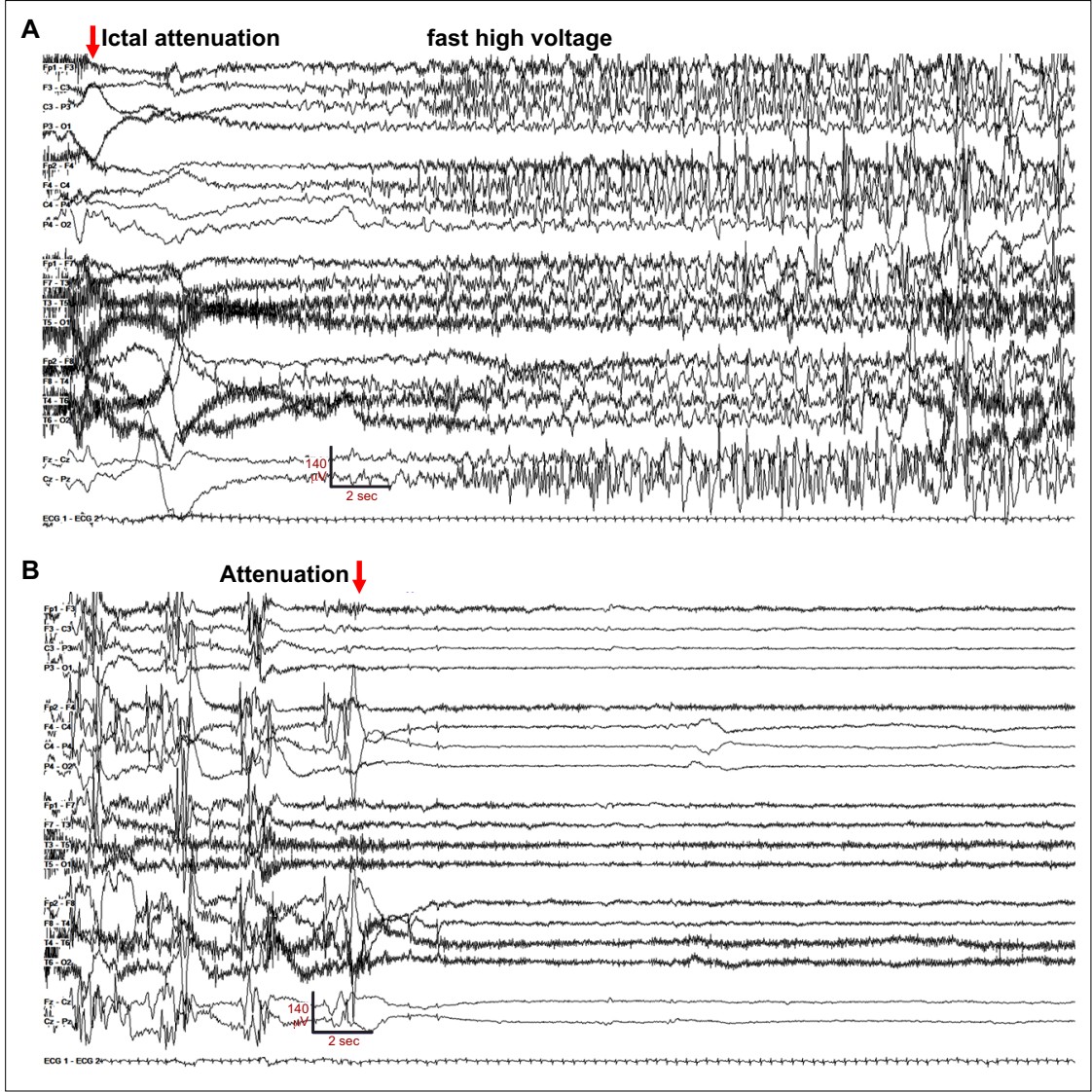

**Figure 1.** EEG of a bilateral onset seizure in G256W/+ individual 1, age 16 days. The recording is continuous from (**A**) to (**B**). (**A**) Seizure onset with diffuse, bilateral amplitude EEG attenuation (red arrow), which is obscured in several electrodes by high-frequency muscle artifact (muscle artifact is better seen in *Figure 1—video 1*). (**B**) Seizure electrographic evolution to post-ictal voltage attenuation (red arrow). Settings: LFF 3 Hz, HFF 70 Hz, sensitivity 7uV/mm, 35 s/panel.

The online version of this article includes the following video and figure supplement(s) for figure 1:

**Figure supplement 1.** Examples of awake and sleep EEG background.

**Figure 1—video 1.** EEG recording including pre-ictal, post-ictal attenuation and recovery of background of seizure excerpted in *Figure 1*.
https://elifesciences.org/articles/91204/figures#fig1video1

The neonate was transferred to a tertiary care hospital for further evaluation and management. Chromosomal microarray, mitochondrial genome, CSF metabolic studies, and sequencing of several individual genes (*ARX, STXBP1, PDHA1, SCN1A*) revealed no abnormalities. Despite antiseizure medication escalations, frequent generalized tonic and focal tonic seizures continued. Review of 158 hr of video-electroencephalography (V-EEG), obtained between age 5 and 21 days, showed 73 seizures where onset, evolution, and offset were individually discernable. There were 2 periods, lasting 15 and 24 min, where several seizures overlapped. Of the well-resolved seizure onsets, 31 (42.4%) were diffuse and bilateral; 42 (57.6%) were unilateral (20 left-sided). *Figure 1* illustrates a seizure of bilateral onset where the initial ictal change was diffuse voltage attenuation (*Figure 1A*, red arrow), a feature described previously (*Ronen et al., 1993*). Marked postictal voltage amplitude attenuation followed 67 of 73 seizures (91.8%, *Figure 1B*; *Figure 1—video 1*). Each period of postictal attenuation was followed by recovery periods with increased discontinuity lasting for up to 5 min after seizures, especially generalized ones (*Figure 1—video 1*). The interictal EEG background was abnormal due to multifocal spikes, poor organization, and excessive discontinuity for age. Unlike in Ohtahara syndrome, however, the EEG showed sleep state dependence, and included some variability and periods of continuity (*Figure 1—figure supplement 1*). The infant was discharged from the hospital on (mg/kg/day) phenobarbital (1.6), levetiracetam (51) and topiramate (9.3), still having multiple daily tonic seizures. At home (age four weeks), new seizure types emerged including myoclonic and epileptic spasms, and additional EEG monitoring showed electrodecrements and multifocal epileptiform activity. Although hypsarrhythmia was not seen, treatment with adrenocorticotropic hormone (ACTH, 120 units/m$^2$/day) was attempted at 1 month of age. This was accompanied by reduced seizure frequency, followed by cessation of clinical seizures within several days. Sanger sequencing of *KCNQ2* at 2 months of age identified a novel heterozygous variant, c.766G>T (p.Gly256Trp), classified as of uncertain clinical significance. Parental testing was not performed. Beginning at 3 months of age, the three antiseizure medicines were reduced sequentially and stopped. The infant remained seizure-free and completed tapering by eight months of age. Over the next dozen years, she experienced infrequent convulsions (6 in total), some provoked by febrile illness or overnight travel.

Developmental delay was clinically apparent in infancy. Assessed by her neurologist at 9 months of age, she was functioning at a level expected typically at 5–6 months. She sat at 1 year, self-fed at 18 months, crawled at 20 months, and walked independently at 42 months. She received diagnoses of cortical visual impairment (1 year) and autism spectrum disorder (3 years). At last examination (age 12), she could use an adaptive communication device to request things ('eat sandwich') and use modified sign language in 2–3 word combinations ('different music please'). She could respond to motor instructions, tap people when she wanted something, and point to items in a book. She is not able to manipulate clothing fasteners or descend stairs without supervision. Developmental challenges have included behavioral outbursts, self-injury, severe constipation, and sleep.

At 15 months of age, parents and physicians began a trial of ezogabine for potential beneficial effects on development. Within 2 weeks the parental global impression was of improved development, but no formal assessment was performed. An attempt to wean and discontinue ezogabine was made after the United States Food and Drug Administration published a notice warning of potential ezogabine-induced risks of skin discoloration, retinal abnormalities, and vision loss (*FDA, U.S, 2013*). Subsequent worsening of irritability, insomnia, and developmental skills with less ezogabine led caregivers to resume the prior dose. When the manufacturer subsequently announced plans for ezogabine market withdrawal, the drug was more slowly tapered and discontinued. A dilated ophthalmologic examination prior to ezogabine discontinuation was normal (29-month exposure).

## Individuals 2-4 with mosaic and heterozygous KCNQ2 G256W

A Korean collaborative team (*Jang et al., 2019*) described Individual 2, a child with neonatal seizures beginning on the second day of life and de novo *KCNQ2* c.766G>T (p.Gly256Trp). The seizure types included focal clonic, tonic, and epileptic spasms. When last seen at age 0.8 years, the child's diagnosis was Ohtahara syndrome, but no additional information was presented. Very recently, a G256W occurrence was identified in Germany (*NCBI, 2023*). The proband male child (Individual 3) experienced seizures beginning at day of life 3 that remitted with antiseizure medicine. Medication was discontinued at one year of age but was resumed in childhood for infrequent convulsions. At last contact (age 12 years), he had delay in fine motor, gross motor, and language development, stereotypies, and

was diagnosed with autism. One parent (Individual 4) had a history of seizures confined to the infantile period, never received antiseizure medication, and experienced subsequent normal development. Sequencing of Individual 3 at age 21 months revealed *KCNQ2* c.766G>T (p.Gly256Trp) heterozygosity. The parent's test showed the same variant in 320 of 831 reads (38.5%), indicative of post-zygotic mosaicism (*Weckhuysen et al., 2012*; *Myers et al., 2018*). These G256W recurrences in individuals manifesting characteristic *KCNQ2* DEE findings provide clinical evidence further supporting pathogenicity. The clinical information highlights electroclinical features we have modelled experimentally.

## G256W lies atop a dome-shaped hydrogen bond network linking helix S5 to the turret and selectivity filter

The KCNQ2 pore domain contains many known pathogenic missense variants and very few non-pathogenic missense variants, leading to low regional mutational tolerance (*Traynelis et al., 2017*; *Karczewski et al., 2020*). G256 lies near the beginning of the loop between the S5 and S6 transmembrane helices (*Figure 2A*). Seeking structural insights into pathogenic mechanisms of the heterozygous (i.e. G256W/+) substitution, we analyzed P-loop regions of cryoelectron microscopic structures from KCNQ2 and several homologues (*Sun and MacKinnon, 2020*; *Li et al., 2021*; *Zheng et al., 2022*). Like other K$^+$ channels, the KCNQ2 P-loop has three distinct subsegments: the turret, the pore helix that partially penetrates the membrane, and the selectivity filter (SF, *Figure 2A–E*). Canonical SF residues T(I/V)GYG line the ion path perpendicular to the membrane surface. The SF polypeptide bends to parallel the membrane, forming a segment (herein termed the SF bridge) that extends radially from the ion path to S6 (*Doyle et al., 1998*; *Hoshi and Armstrong, 2013*). G256 faces the extracellular aqueous environment at the periphery and apex of the PGD, over 22 Å away from the SF (*Figure 2E–F*, *Figure 2—video 1*). We wondered how substitution at G256 might alter pore function. We compared structural models of KCNQ2, KCNQ4, and KCNQ1 with those of the more distantly related channels KcsA, fly *Shaker,* and human Kv1.2 and made phylogenetic sequence comparisons (*Figure 2—figure supplement 1*). Unlike in KCNQ1, KcsA, *Shaker*, or Kv1.2, KCNQ2 structural models and their density maps revealed G256 as the apical node of an H-bond network arching from S5 to the KCNQ2 SFB segment (*Figure 2—video 2*, *Figure 2G–I*, *Figure 2—figure supplement 1*). Glycine uniquely confers main chain flexibility due to its lack of steric hindrance (*Carugo and Djinovic-Carugo, 2013*). KCNQ2 G256 contributes to a tight peptide turn through torsion angles (psi –64.4° and phi –74.3°) rarely found at non-Gly residues. The model predicts a Gly256-Glu257 $\omega$-bond deviates from planarity, which is unusual (*MacArthur and Thornton, 1996*), contributing to a G256 carbonyl to N258 amide H-bond (*Figure 2G–I*). The N258 side chain extends away from G256 towards the SFB, where it contributes to a network of bonds including three turret (N258, H260 and D262) and three SFB residues (K283, Y284, and Q286).

Phylogenetic comparisons indicate that four turret and SFB residues (G256, H260, D262, and Q286) within this bond network co-evolved in *KCNQ2* genes of fish and amphibians, and were subsequently conserved across reptiles, birds, and mammals, divergent from other *KCNQ* subtypes (*Figure 2C*). KCNQ4, an evolutionary ancestor of both KCNQ2 and KCNQ3 (*Cooper, 2011*), exhibits a very similar turret fold and a bond network linking S5, turret, and SFB (*Figure 2—figure supplement 1C, D*). Turret sequences and structures of KCNQ1, KcsA, *Shaker,* and Kv1.2 are less like KCNQ2 and lack direct bonds to the SFB. The turret sequence of KCNQ3, the most recently arising KCNQ2 paralogue, includes 10 inserted residues that are absent from KCNQ5, 4, and 2 (*Figure 2—figure supplement 1F*). This evidence of unique divergence and later conservation suggested that despite its small side chain, water-facing location, and distance from the pore, KCNQ2 G256 could be intolerant to substitution. Beyond the loss of unique Gly physicochemical features, introducing a large, planar, hydrophobic W256 side chain might cause disruptive local consequences due to its preference for burial at membrane boundaries (*Khemaissa et al., 2022*). We tested these predictions with experiments in vitro and using mice bearing the heterozygous G256W substitution.

## G256W co-expression suppresses currents of KCNQ2/KCNQ3 channels in Chinese hamster ovary (CHO) cells

We made whole cell patch clamp recordings from CHO cells co-expressing wild-type (WT) KCNQ2 and KCNQ3 using an automated, 384-well system (*Vanoye et al., 2022*). We compared three expression conditions: WT KCNQ2 and KCNQ3, G256W and WT KCNQ3, and co-expression of a 1:1 ratio of

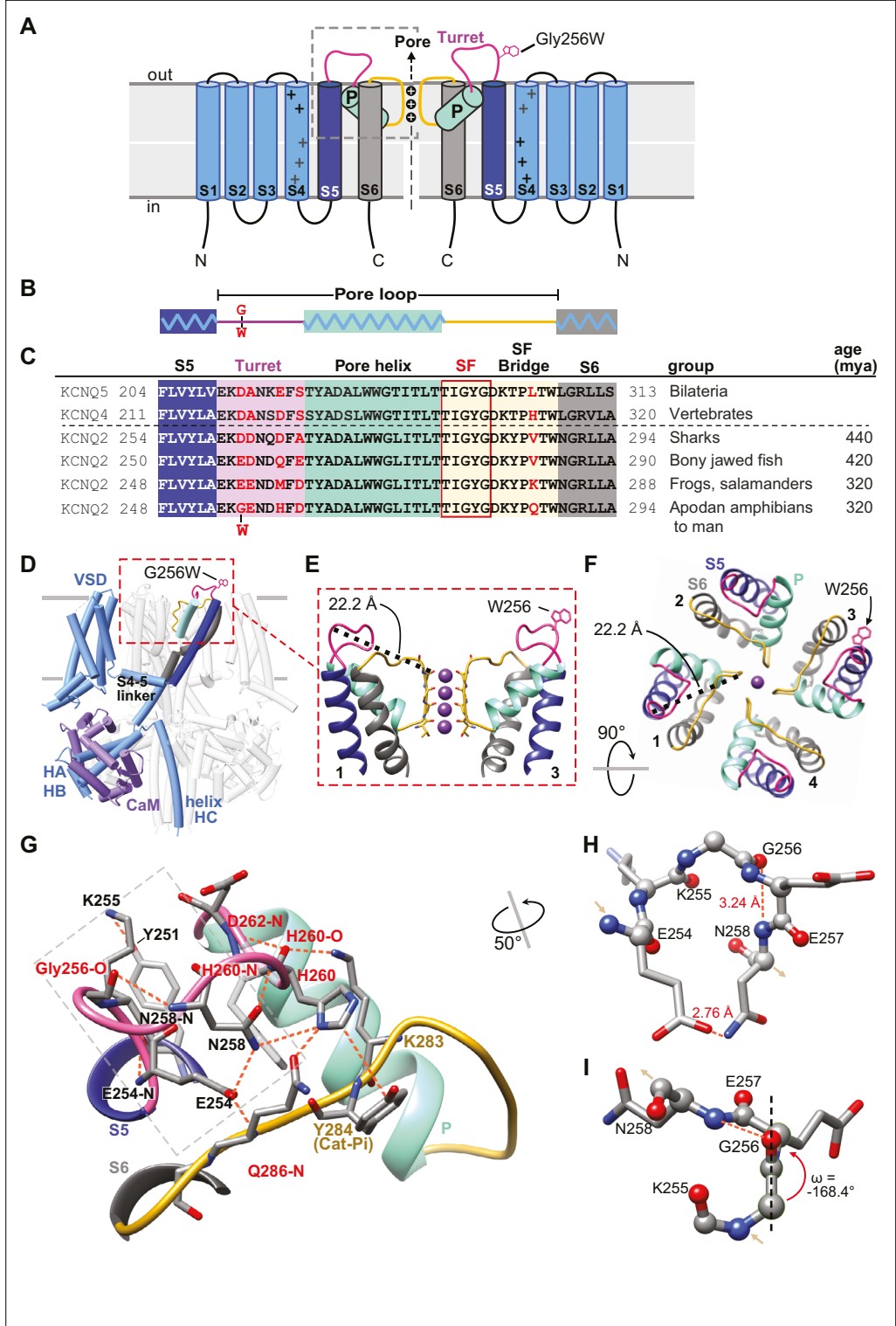

**Figure 2.** Gly256 is linked to the selectivity filter bridge segment via a hydrogen bond network among residues distinct to KCNQ2. (**A, B**) Cartoons showing KCNQ2 membrane topology, including transmembrane segments S1-S6 and the P-loop (turret segment, purple; H5 or P-helix, cyan; and selectivity filter segment, yellow). Positions of the K⁺ selective pore, and the G256W substitution within the turret are indicated. (**C**) Alignment of human KCNQ4 and KCNQ5 sequences with KCNQ2 sequences of major vertebrate groups. Background colors match panels A-B, and the five selectivity filter lining residues are boxed in red. At four aligned positions within the turret and one in the SFB, KCNQ2 substitutions have evolved in amphibians and tetrapods (residues highlighted in red).

*Figure 2 continued on next page*

*Figure 2 continued*

(**D**) Rendering of the WT KCNQ2-calmodulin tetrameric structure obtained by cryoEM (PDB 7cr3), highlighting one subunit and the position of the G256W substitution near the channel's extracellular domain apex. The Trp256 sidechain is at scale but its rotamer is chosen arbitrarily. The subunit closest to the viewer is partially deleted to reveal the highlighted subunit more clearly. (**E**) Ribbon rendering of the extracellular part of the PGD. For clarity, only two opposing side subunits are shown (as schematically in **A**). A Trp side chain is added at one Gly256 α-carbon. The distance between the G256 α-carbon and Y280 carbonyl oxygen at the selectivity filter mouth is labeled. (**F**) Top down view of the KCNQ2 regions as in panel (**E**) but showing four subunits. The Trp rotamer is different from panels (**D, E**) The S5, S6, and P-helices are labeled. (**G**) Hydrogen bonding network of the KCNQ2 turret. All predicted bonds are shown as dashed orange lines. The network extends from the S5 helix (Y251) via the labelled turret residue atoms to bonds involving residues of the SFB. As in (**C**), KCNQ2 residues that diverge in vertebrates are colored red. (**H, I**) The turret peptide region near G256, which is boxed with a grey dashed line in (**G**). The main chain is shown as ball-and-stick; side chains as stick. A tight turn occurs at K255 to N258, stabilized by hydrogen bonding between the G256 carbonyl oxygen and N258 amide. (**I**) The G256-E257 peptide deviates from planarity ($\omega$ = +/-180°) by 11.6° (~2.6 sd). In and out arrows indicate N and C termini, respectively. Abbreviations: mya, million years ago; VSD, voltage-sensor domain; HA-HB, the cytoplasmic helices A and B; CaM, calmodulin.

The online version of this article includes the following video and figure supplement(s) for figure 2:

**Figure supplement 1.** The G256W variant affects a divergent neuronal KCNQ turret structure enabling forming a bonding network linked to the ion selective pore.

**Figure 2—video 1.** Movie illustrating position of the G256W substitution within the KCNQ2 channel pore turret and its distance to the selectivity filter.

https://elifesciences.org/articles/91204/figures#fig2video1

**Figure 2—video 2.** Movie illustrating locations of residues contributing to a non-covalent bonding network extending from S5 to the selectivity filter.

https://elifesciences.org/articles/91204/figures#fig2video2

---

WT KCNQ2 and G256W with WT KCNQ3 to mimic the heterozygous genotype. A simplified random association model assuming equal expression and assembly predicts that the last condition results in a mixed population of subunit stoichiometries containing 0, 1, or 2 G256W subunits (*Figure 3A*). The WT channels gave currents with slow voltage-dependent activation and no inactivation (*Figure 3B*). Cells expressing either KCNQ3 only (*Vanoye et al., 2022*), or the combination of KCNQ2 G256W and WT KCNQ3 (*Figure 3B*) exhibited no detectable currents. When WT KCNQ2 and G256W cDNAs were co-expressed at a 1:1 ratio to mimic heterozygosity, currents at 40 mV were significantly reduced to 44.3 ± 8% compared to WT only controls (*Figure 3B and C*). This reduction appeared linear with respect to time and voltage, as conductance-voltage fits showed no significant changes in $V_{1/2}$ (*Figure 3E*), activation slope, or time constants.

Ezogabine (retigabine) increases currents through neuronal KCNQ channels by shifting activation to more hyperpolarized voltages, enhancing activation kinetics, and increasing maximal current density (*Tatulian et al., 2001*; *Gunthorpe et al., 2012*). During its period of commercial availability, ezogabine was used as targeted therapy in patients with *KCNQ2* DEE arising from loss-of-function variants (*Millichap and Cooper, 2012*; *Weckhuysen et al., 2013*; *Millichap et al., 2016*; *Nissenkorn et al., 2021*), including in individual 1. We compared the effects of ezogabine (10μM) on WT KCNQ2/KCNQ3 heteromers and channels from cells expressing G256W. In cells expressing G256W and KCNQ3, ezogabine treatment had no effect (*Figure 3C and F*). In cells expressing WT KCNQ2, G256W and WT KCNQ3, mimicking heterozygosity and predicted to yield a mixed population of channels (*Figure 3A*), ezogabine significantly increased currents and shifted activation voltage dependence (*Figure 3C, D and E*). Currents from the heterozygous G256W condition after ezogabine exceeded those of WT KCNQ2/KCNQ3 control cells absent ezogabine. We compared the relative effect of ezogabine in control and G256W/+ cells by dividing post- by pre-treatment currents. Ezogabine enhancement in cells mimicking the G256W/+ genotype was equal to that of WT controls (*Figure 3G*). This is especially notable, as 25% of the channels assembled in the G256W/+ mimicking cells are predicted to include two G256W subunits and therefore be ezogabine-unresponsive. This suggested ezogabine treatment increased current in cells including one G256W subunit as fully as in WT KCNQ2/KCNQ3 channels.

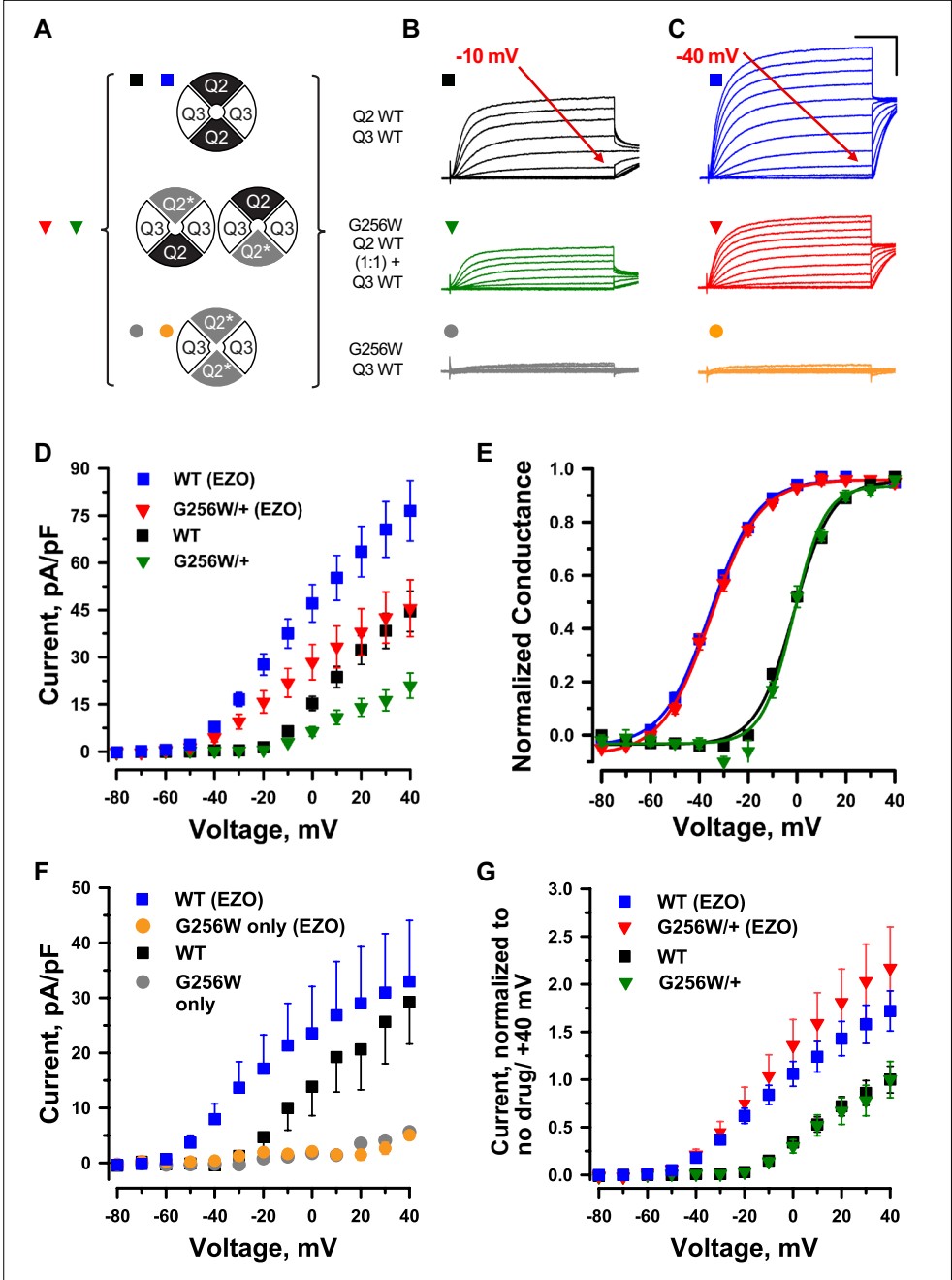

**Figure 3.** KCNQ2 G256W co-expression suppresses current in KCNQ2/KCNQ3 heteromeric channels. (**A**) Cartoon showing the expected combinations of WT and G256W subunits under heterozygosity based on a simple random association model and preferred 2:2 stoichiometry for KCNQ2 and KCNQ3. (**B–G**) In vitro dissection of effects of G256W heterozygosity on currents. (**B, C**) Mean current families are shown for the indicated combinations of expression of KCNQ2 and KCNQ3 prior to and after addition of 10 μM ezogabine (n=60, 50; 40, 31; 28, 24 for the upper, middle, and lower conditions). Scale: 250ms and 20 pA/pF. (**D, E**) Current/voltage and conductance/voltage relationships for the indicated WT only and G256W/WT electroporations into KCNQ3 stable expressing cells. (**F**) Current/voltage relationship for G256W ('homozygous') heteromeric channels, compared with the subset of WT control cells studied in parallel by automated patch recording. (**G**) Replot of data from panel (**D**) At each voltage, mean current is normalized to mean current at +40 mV in absence of ezogabine.

The online version of this article includes the following source data and figure supplement(s) for figure 3:

**Source data 1.** Mean numerical data, n's, and mutagenic primers used to produce *Figure 3*.

*Figure 3 continued on next page*

*Figure 3 continued*

**Figure supplement 1.** KCNQ2 G256W co-expression suppresses current in KCNQ2/KCNQ3 heteromeric channels recorded by manual patch-clamp.

**Figure supplement 1—source data 1.** Numerical data and calculations used to produce *Figure 3—figure supplement 1*.

**Figure supplement 2.** KCNQ2 G256W co-expression suppresses current in KCNQ2 homomeric channels recorded by manual patch-clamp.

**Figure supplement 2—source data 1.** Numerical data and calculations used to produce *Figure 3—figure supplement 2*.

## G256W co-expression suppresses homomeric KCNQ2 currents in Chinese hamster ovary (CHO) cells

In parallel, we studied the effect of G256W through manual whole-cell patch recordings. We used a co-transfected GFP marker to select for high expression cells for patching. Mean current densities were higher than observed for the automated patch recording. However, as in the automated patch system, co-expression of G256W with WT KCNQ3 resulted in little or no current. Co-transfection of G256W, WT *KCNQ2* and *KCNQ3* cDNA (1:1:2) to mimic heterozygosity reduced current to 41%+/-0.1% of WT controls (*Figure 3—figure supplement 1*). Because KCNQ2 subunits appear to be expressed as KCNQ2 homomeric channels in some neurons (*Cooper et al., 2000*; *Hadley et al., 2003*; *Martire et al., 2004*; *Schwarz et al., 2006*; *Varghese et al., 2023*), we analyzed the impact of G256W in this subunit configuration. WT KCNQ2 expressed alone gave robust currents, but G256W currents were undetectable (*Figure 3—figure supplement 2*). Co-expression of WT KCNQ2 and G256W cDNA (1:1) has been modeled as yielding a mix of tetramers including WT subunits only, G256W only, and between one and three G256W subunits, with the ratios of these stoichiometries predicted by the binomial distribution (*Figure 3—figure supplement 2D*). The current density (+40 mV) under 1:1 co-expression to mimic heterozygosity was 30.1%+/-0.056% of WT control, a greater relative reduction than seen for heteromeric channels, as predicted by with the assembly models. Heterozygous co-expression of G256W did not change the midpoint voltage or steepness of voltage-dependent activation in either the heteromeric or homomeric subunit composition experiments.

## A neonatal seizure in a heterozygous G256W mouse at P10

To better understand the consequences of the G256W/+ expression in vivo under control of the native *Kcnq2* promoter, we introduced the variant into C57BL/6J mice using Crispr/Cas9. Initial progeny included mice heterozygous for the intended variant (G256W/+) and a founder mouse heterozygous for a 7 base deletion in codons 254–256 (E254fs; *Figure 4A*). The frameshift deletion preserved the WT exon 5–6 splice boundary, yielding a transcript with 16 novel sense codons and an in-frame stop codon spanning the exon 5–6 junction (*Figure 4—figure supplement 1*). The transcript is predicted to yield a truncated protein and be targeted for nonsense mediated mRNA decay. Western blots of brain tissue from *Kcnq2*[E254fs/+] mice using antibodies against the KCNQ2 N-terminus did not reveal bands for the protein product (*Figure 4—figure supplement 2*). Truncating variants are found in a majority of SLFNE families (59.5%; *Goto et al., 2019*). To enable direct comparison between a characteristic SLFNE variant type and G256W in vivo, we purified both lines by breeding against WT C57BL/6J and studied them in parallel.

In crosses of WT females and heterozygous males, G256W/+ and E254fs/+ mice represented ~50% of live births (230/481 and 147/309, respectively). For E254fs het x het crosses, 12 WT, 33 E254fs/+, and 3 E254fs/E254fs mice were born. The E254fs/E254fs pups appeared stillborn or died at P0, as seen for homozygotes with other *Kcnq2* null alleles (*Watanabe et al., 2000*; *Yang et al., 2003*). For G256W het x het crosses, 16 WT, 24 G256W/+, and 5 G256W/G256W mice were born. All G256W/G256W pups were dead at initial observation on P0 or died within hours. G256W/+ pups showed no differences in weight nor in the screens for progress in several assays of motor development (*Figure 4B and C*). However, we video recorded a convulsive seizure in a P10 G256W/+ female (*Figure 4D–H*). The seizure evolved from behavioral arrest and myoclonic jerks, followed by loss of postural control and hindlimb/tail extensor posturing. After 90 s of immobility with brief episodes of myoclonus, the mouse regained awareness and upright posture. The seizure lasted approximately

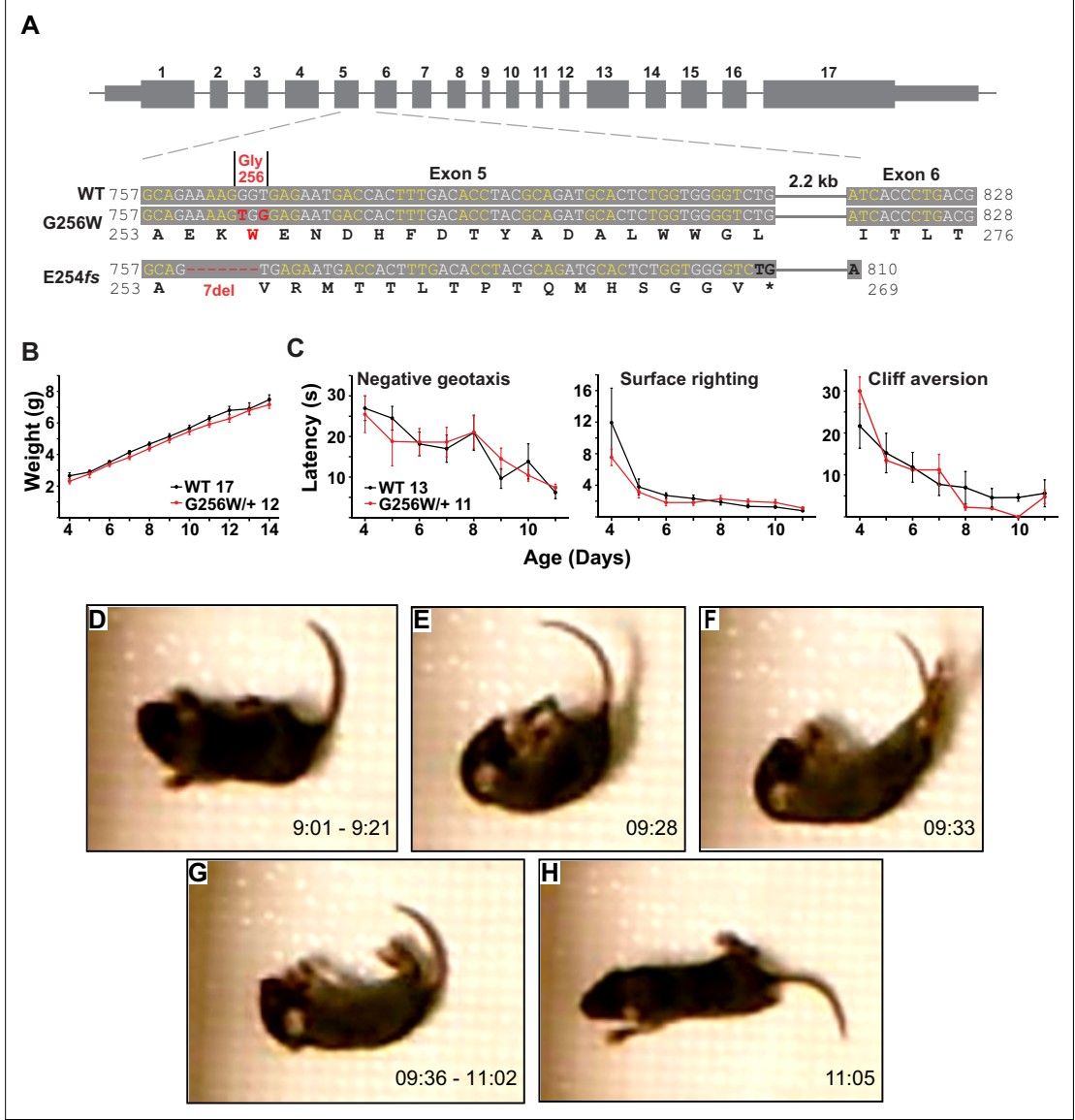

**Figure 4.** Immature heterozygous G256W mice exhibit normal development and have infrequent epileptic seizures. (**A**) Upper, map of the *Kcnq2* constructs. Lower, sequence alignments for the region between the middle of exon 5 and the beginning of exon 6. Although the human G256W variant is a single base substitution, Crispr/Cas9 editing introduced two substitutions, since the WT G256 codons differ between mouse (GGT) and human (GGG). Also aligned is the DNA and protein sequences of the frameshift mutation. (**B**) WT and G256W/+ mice showed no difference in weight gain during development. (**C**) WT and G256W/+ mice performed similarly in the developmental milestone assays for negative geotaxis, surface righting, and cliff aversion. (**D–H**) Screenshots of stages of a generalized seizure in a P10 G256W/+ mouse (see also: *Figure 4—video 1*). (**D**) Onset with immobility and myoclonic tail and forelimb shaking. (**E**) Abrupt fall to side with flexion posturing. (**F**) Evolution to hindlimb and tail extension posture. (**G**) Immobility with flaccid appearance, interrupted by brief episodes of tail, individual limb myoclonus or clonus. (**H**) Arouses, quickly regains upright posture, then normal mobility. Labels: time in 15 min source video.

The online version of this article includes the following video, source data, and figure supplement(s) for figure 4:

**Source data 1.** Individual mouse and mean data used for *Figure 4B*.

**Figure supplement 1.** DNA, RNA, and predicted protein consequences of the G256W and E254fs*16 mutations.

**Figure supplement 2.** Western blotting reveals no evidence of the predicted E254fs truncated protein product.

**Figure supplement 2—source data 1.** Band signal intensity data for the 28 kD band graphed in *Figure 4—figure supplement 2C*.

**Figure supplement 2—source data 2.** This file is a zip folder containing an image of the uncropped blot with the relevant bands clearly labelled seen in *Figure 4—figure supplement 2A, B*.

**Figure supplement 2—source data 3.** This file is a zip folder containing the original files of the full raw unedited blot seen in *Figure 4—figure*

*Figure 4 continued on next page*

*Figure 4 continued*

*supplement 2A, B*.

**Figure 4—video 1.** Generalized seizure in a P10 heterozygous G256W mouse.
https://elifesciences.org/articles/91204/figures#fig4video1

120 s in total (*Figure 4—video 1*). Between five and nine G256W/+ mice and 26 control mice were video recorded on alternate days between P8 and P14 for 15 min per day, and only one seizure was observed.

## CA1 pyramidal neurons in P12-15 heterozygous G256W mice show increased firing and reduced spike frequency adaptation

We made whole cell recordings of CA1 pyramidal cells in acute horizontal hippocampal slices from P12-15 WT and G256W/+ littermates. Compared to WT neurons, positive current injections in G256W/+ neurons evoked significantly greater numbers of action potentials (*Figure 5*). There were no differences in resting membrane potential, several action potential biophysical parameters, or somatic input resistance between WT and G256W/+ mice (*Figure 5—figure supplement 1*).

## Adult heterozygous G256W mice experience fatal and nonfatal seizures

Convulsive seizures were observed in adult G256W/+ mice occasionally during routine animal care but never in co-housed WT littermates. To learn electrophysiological correlates, we performed

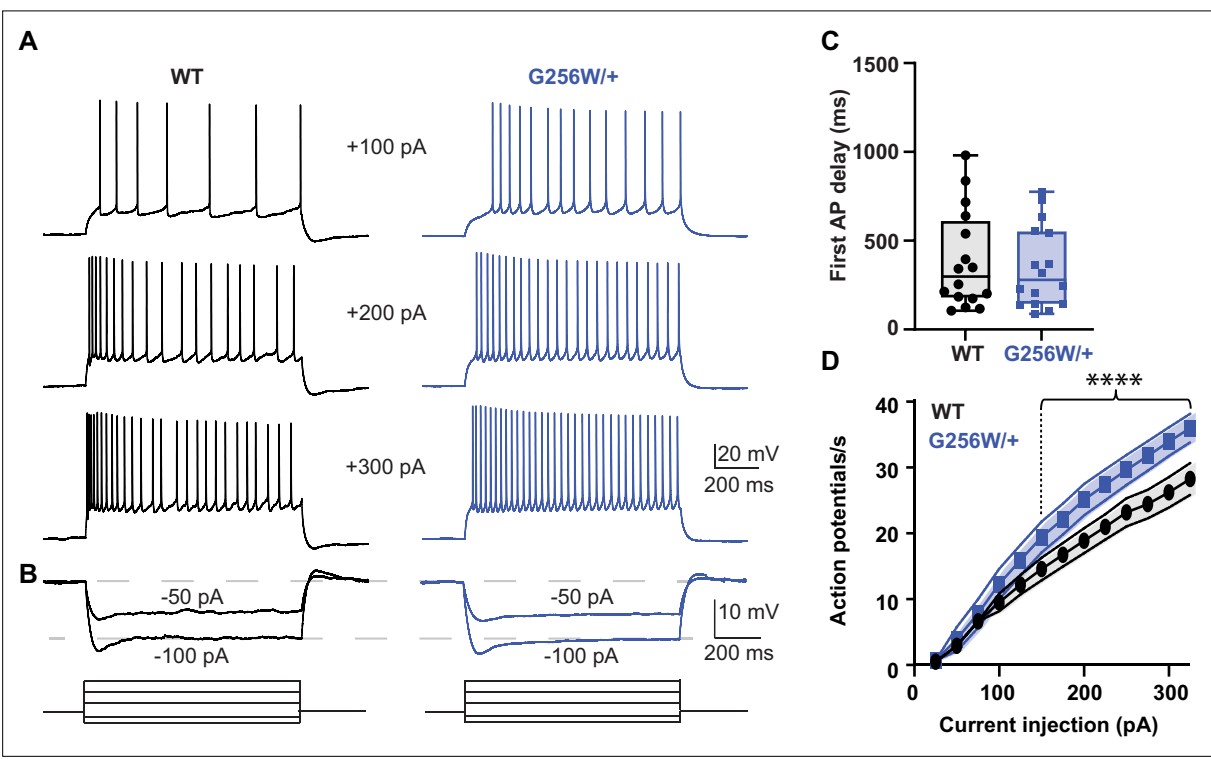

**Figure 5.** Heterozygous G256W mice have increased CA1 pyramidal cell excitability. (**A**) Representative voltage responses to increasing current injection steps (step duration 1 s) in CA1 pyramidal neurons from WT and G256W/+ mice. The resting membrane potential was held at −65 mV. (**B**) Representative voltage responses to decreasing current injections steps (1 s) in CA1 pyramidal neurons from WT and G256W/+ mice. (**C**) Time to first action potential following step stimulus is not significantly different between groups (3 animals per group; WT and G256W/+, n=16 cells each). (**D**) Summary graph showing the effect of one copy of G256W on the action potential count (3 animals per group; WT and G256W/+, n=16 cells each, F(12,180)=5.8, **** is p<0.0001). Data are presented as mean and s.e.m.

The online version of this article includes the following source data and figure supplement(s) for figure 5:

**Figure supplement 1.** Several neuronal biophysical properties are unchanged.

**Figure supplement 1—source data 1.** Individual cell data used for *Figure 5*, *Figure 5—figure supplement 1*.

electrocorticography on eight adult heterozygous mice for a total of 1440 hr. The five electrographic seizures captured in two mice were stereotyped. All exhibited generalized onset and stereotyped evolution: an isolated herald spike or polyspike (*Figure 6A and B*, arrows), rapid transition to fast, high-amplitude spiking lasting 20–30 s, strong amplitude attenuation lasting 15–30 s, and slow recovery (*Figure 6A*). One mouse was recorded for 150 hr over 10 days without seizures, then had four seizures within 72 hr, the last of which was fatal. The final ictal EEG followed the pattern of prior seizures, but the strongly attenuated EEG diminished progressively and did not recover (*Figure 6B and C*). One fatal seizure was video recorded (*Figure 6—video 1*). In this mouse, seizure began with wild running and jumping (for 5–10 s), followed by a brief arrest (and start of the video recording). Running resumed for 3 s, followed by abrupt loss of postural control, extensor posturing, and respiratory arrest. The experimenter attempted cardiac resuscitation with anterior chest compressions, but the mouse did not revive. In other seizure instances, mice were observed to revive, either without intervention or with chest compressions or a noxious limb pinch stimulus. Kaplan-Meier analysis showed that ~22% of non-censored G256W/+ mice died prematurely, a significant increase compared to WT littermates, with a median age of death of 180 days (*Figure 6D*). E254fs/+ mice experienced no early mortality, in agreement with findings for other SLFNE models (*Watanabe et al., 2000*; *Singh et al., 2008*; *Robbins et al., 2013*; *Figure 6—figure supplement 1*). Many G256W/+ premature deaths during long-term colony housing were unwitnessed, but the mouse bodies we recovered after death exhibited the stereotyped posture we observed in the two seizure videos, with flexed forelimb and hindlimb extension posture captured by rigor mortis.

## Distinct patterns of altered *Kcnq2* and *Kcnq3* mRNA expression in heterozygous E254fs and G256W mice

We used RT-qPCR to measure *Kcnq2* and *Kcnq3* mRNA levels in hippocampus and neocortex at P21 and P100. There were no differences in *Kcnq2* mRNA expression between WT and G256W/+ mice (*Figure 7A*). Because E254fs transcripts should be eliminated by nonsense mediated decay, we expected E254fs/+ mice to have about 50% less *Kcnq2* mRNA than WT and G256W/+ mice. However, using an assay that did not discriminate between transcripts from WT and E254fs alleles, *Kcnq2* mRNA was reduced by only 25% (+/-3%) at P21 and by 35% (+/-1%) at P100 in E254fs/+ mice. The *Kcnq2* mRNA levels in E254fs/+ mice were significantly greater than 50% of WT (one sample t-test). This result could reflect incompletely efficient nonsense mediated decay of E254fs transcripts (*Zetoune et al., 2008*; *Dyle et al., 2020*), or compensation achieved via increased *Kcnq2* gene expression or mRNA stability. To attempt to discriminate among those possibilities, we performed additional experiments using a TaqMan assay detecting only the WT *Kcnq2* allele (*McGuigan and Ralston, 2002*). In this assay, E254fs/+ and G256W/+ mice both showed approximately 50% losses of mRNA. In E254fs/+ mice, total *Kcnq2* mRNA was significantly greater than WT *Kcnq2* mRNA (one sample t-test; *Figure 7B*). Taken together, these results suggest that nonsense-mediated decay of E254fs transcripts is incomplete, and that expression of the *Kcnq2* WT allele is unchanged in both lines of heterozygous mice. In contrast, RT-qPCR revealed a significant increase in *Kcnq3* mRNA in E254fs/+ hippocampus at P21, as observed previously for *Kcnq3* mRNA *Kcnq2* homozygous null mouse models (*Robbins et al., 2013*). In addition, *Kcnq3* mRNA was increased in three of four G256W/+ experiments (P21 cortex, P21 hippocampus and P100 hippocampus but not P100 cortex). Consistent with the qPCR results, Sanger sequencing of cDNA amplified from P21 hippocampus of G256W/+ and E254fs/+ mice showed peaks corresponding to the G256W and E254fs alleles (*Figure 7—figure supplement 1B, C*).

## Heterozygous G256W mice show diminished KCNQ2 and KCNQ3 targeting to axon initial segments and axons, and reduced levels of KCNQ2 protein

KCNQ2 is highly concentrated at many axon initial segments and nodes of Ranvier, where it colocalizes with voltage-gated sodium (NaV) channels, Ankyrin-G, and other Ankyrin-G interacting proteins (*Devaux et al., 2004*; *Pan et al., 2006*). In many but not all neurons, KCNQ3 is also colocalized at these subdomains and forms heteromeric channel with KCNQ2 (*Shah et al., 2008*; *Klinger et al., 2011*; *Battefeld et al., 2014*; *Martinello et al., 2015*; *Jing et al., 2022*). We compared KCNQ2 and KCNQ3 subcellular localization in WT, E254fs/+ and G256W/+ mice by immunofluorescence labeling

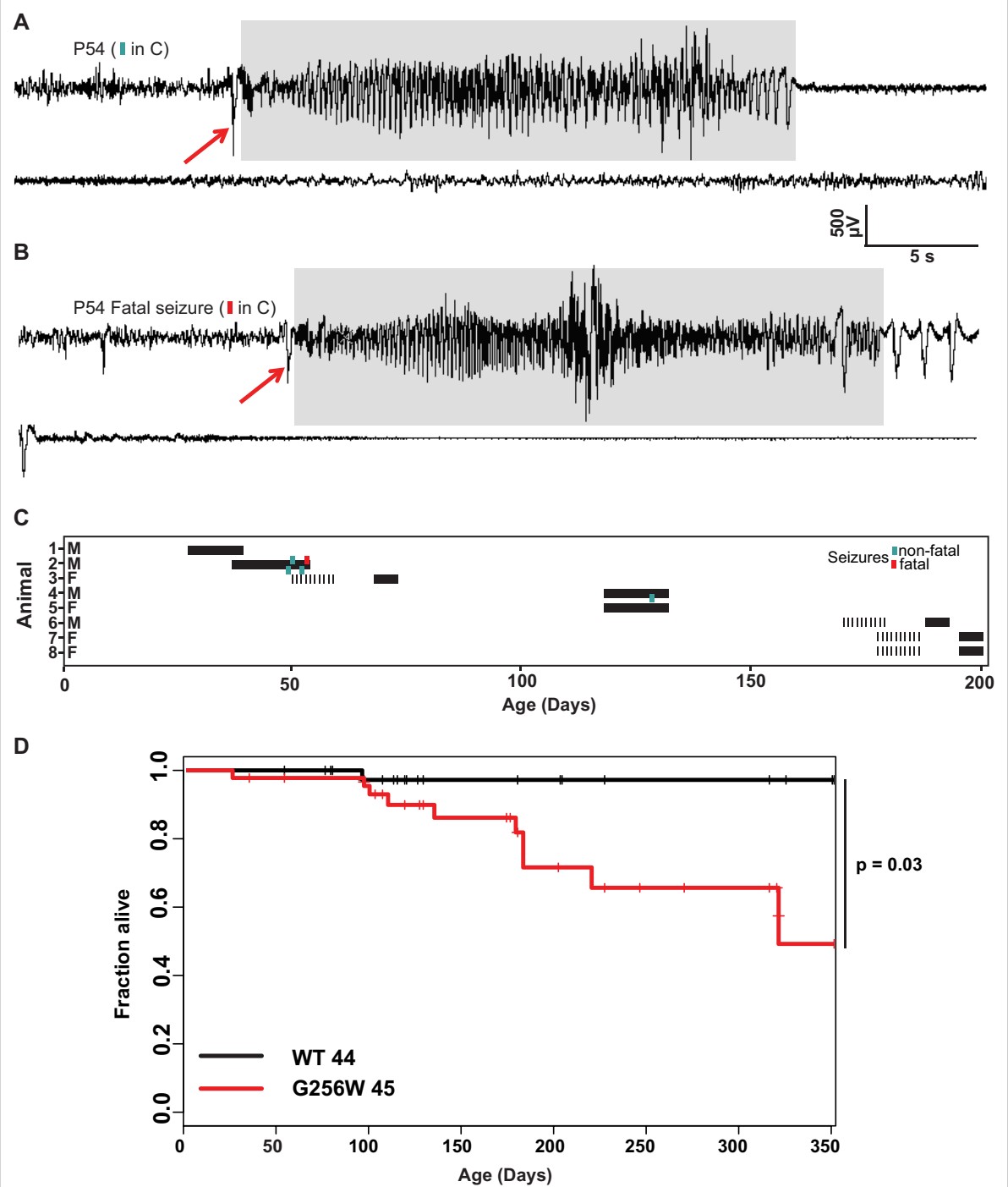

**Figure 6.** Convulsive seizures in adult heterozygous G256W mice show stereotyped electrographic features and reduced survival. (**A, B**) EEGs of non-fatal and subsequent fatal seizure captured in a P54 male G256W/+ mouse (animal 2, **C**). Electrographic seizures were characterized by fast spiking, high amplitude activity lasting 15–20 s (highlighted in gray). Summary showing the sex, ages, duration of recordings and timing of seizures in 8 animals undergoing EEG. Turqoise hashmarks denote a survived seizure, red hashmark denote a fatal seizure. Black bars are periods on EEG; some recording were performed on a 6 hr/day schedule. (**D**) Survival curve of WT vs G256W/+ mice, hashmarks indicate censored mice. G256W/+ mice had signifcant mortality, p=0.0348 Cox propotional hazards model.

The online version of this article includes the following video, source data, and figure supplement(s) for figure 6:

**Source data 1.** Individual mouse survival data used for *Figure 6D*.

**Figure supplement 1.** No significant mortality in heterozygous E254fs mice.

*Figure 6 continued on next page*

*Figure 6 continued*

**Figure supplement 1—source data 1.** Individual mouse survival data used for *Figure 6—figure supplement 1*.

**Figure 6—video 1.** Video of a fatal convulsive seizure in a 4-month-old heterozygous G256W mouse.

https://elifesciences.org/articles/91204/figures#fig6video1

and confocal imaging of hippocampal sections, using antibodies directed against sequences from the KCNQ2 and KCNQ3 N-terminal regions and trios of samples processed and imaged in parallel. In CA1, where abundant pyramidal cell AISs are found crossing the border between stratum pyramidale and stratum oriens (*Video 1*), sections from G256W/+ mice showed diminished AIS labeling and increased neuronal somata labeling for both KCNQ2 and KCNQ3. In the CA1 of E254fs/+ mice, the intensity ratios of AIS and somatic labeled regions did not differ from WT (*Video 2*). In CA3, images of G256W/+ mice showed less intense KCNQ2 and KCNQ3 labeling within stratum lucidum (where mossy fibers are marked by PanNav) and increased labeling in stratum pyramidale (*Video 3*).

To test these results obtained by visual inspection of paired samples, we performed a quantitative, blinded analysis of subcellular labeling of WT, G256W/+, and E254fs/+ mouse hippocampal sections in CA1, CA3, and the dentate gyrus. We marked axonal and soma containing regions of interest (ROIs) and quantitated relative axon (or AIS) to somatic containing ROI intensities. Comparisons between these ratios reflect relative axonal (or AIS) and somatic protein concentrations. The approach does not quantify at the level of individual axons and somata, and linearity of labeling intensity vs. protein was not established. Because these limitations applied equally across genotypes, however, the methods allowed differences between the genotypes to be detected and tested for significance. For G256W/+ mice, the ratio of pyramidal cell AIS/somatic ROI intensity was significantly reduced for both KCNQ2 and KCNQ3 in the CA1 and CA3 (*Figure 8A and B*). In CA3, this was driven mainly by increased pyramidal cell somatic labeling, as the long CA3 AISs (*Kosaka, 1980*) are more sparse than in CA1 and were infrequently captured longitudinally in coronal sections. Also, G256W/+ mice had a significant reduction in the CA3 stratum lucidum to stratum pyramidale ROI ratio, suggesting potential diminished forward trafficking to the mossy fibers and/or CA3 pyramidal cell axon collaterals (*Figure 8C*). The very thin AISs of dentate granule cells lie mostly within the granule cell layer (GCL; *Martinello et al., 2015*), so measurement of ratios between tissue layers is an imperfect proxy for (soma vs. AIS) subcellular distribution. The KCNQ2 labeling ratio for dentate polymorphic layer (PML; axon-enriched) vs. GCL (somata and AISs) was not significant different between genotypes, but the KCNQ3 ratio showed significant reduction (*Figure 8D*).

In the dorsal hippocampal CA1 region, pyramidal cells outnumber interneurons about 100-fold (*Keller et al., 2018*). We did not investigate labeling of the hippocampal interneurons extensively, due to their less frequent appearance in the matched sections used for analysis of excitatory neurons. In six CA1 image stacks of G256W/+ mice, 11 putative interneurons showed conspicuous KCNQ2 somatic labeling (e.g. *Video 1*, arrow). Only one of these 11 interneurons was co-labeled for KCNQ3 (*Figure 8—figure supplement 1*). In an equally sized sample of WT image stacks, we identified 19 likely interneurons; of those, none were somatically labeled for KCNQ2 or KCNQ3. A small subset of interneuron AISs of both genotypes showed weak labeling of for KCNQ2 and KCNQ3.

We quantified KCNQ2 and KCNQ3 protein levels in neocortical tissue from P21 mice by western blot. As channel proteins localized to AISs and somatic domains could potentially track to different biochemical subcellular fractions, we performed preliminary experiments and developed sample preparation methods that allowed protein from whole tissue homogenates to be denatured and loaded onto SDS gels. We omitted centrifugal fractionation steps to avoid loss of detergent-resistant proteins in low-speed pellets, which might differ by genotype. Across conditions tested, western blots revealed the biochemical properties of KCNQ2 and KCNQ3 to be strikingly different, despite the high homology of the two protein sequences. E254fs/+ and G256W/+ samples showed an approximate 50% reduction in KCNQ2 monomer band versus WT, but no change in KCNQ3 (*Figure 9*). KCNQ2 blots showed additional bands corresponding to candidate dimer and higher oligomeric forms, but KCNQ3 blots showed a predominant monomer (*Figure 9—figure supplement 1*). KCNQ2 monomer bands showed microheterogeneity, perhaps reflecting expression of multiple splice isoforms (*Cooper et al., 2001*; *Pan et al., 2001*). All KCNQ3 blots showed a single band running near the ~97 kD mass calculated from its cDNA. To estimate total KCNQ2 protein, we quantified higher molecular weight KCNQ2 bands and the summed intensity of monomeric and higher molecular weight KCNQ2 bands.

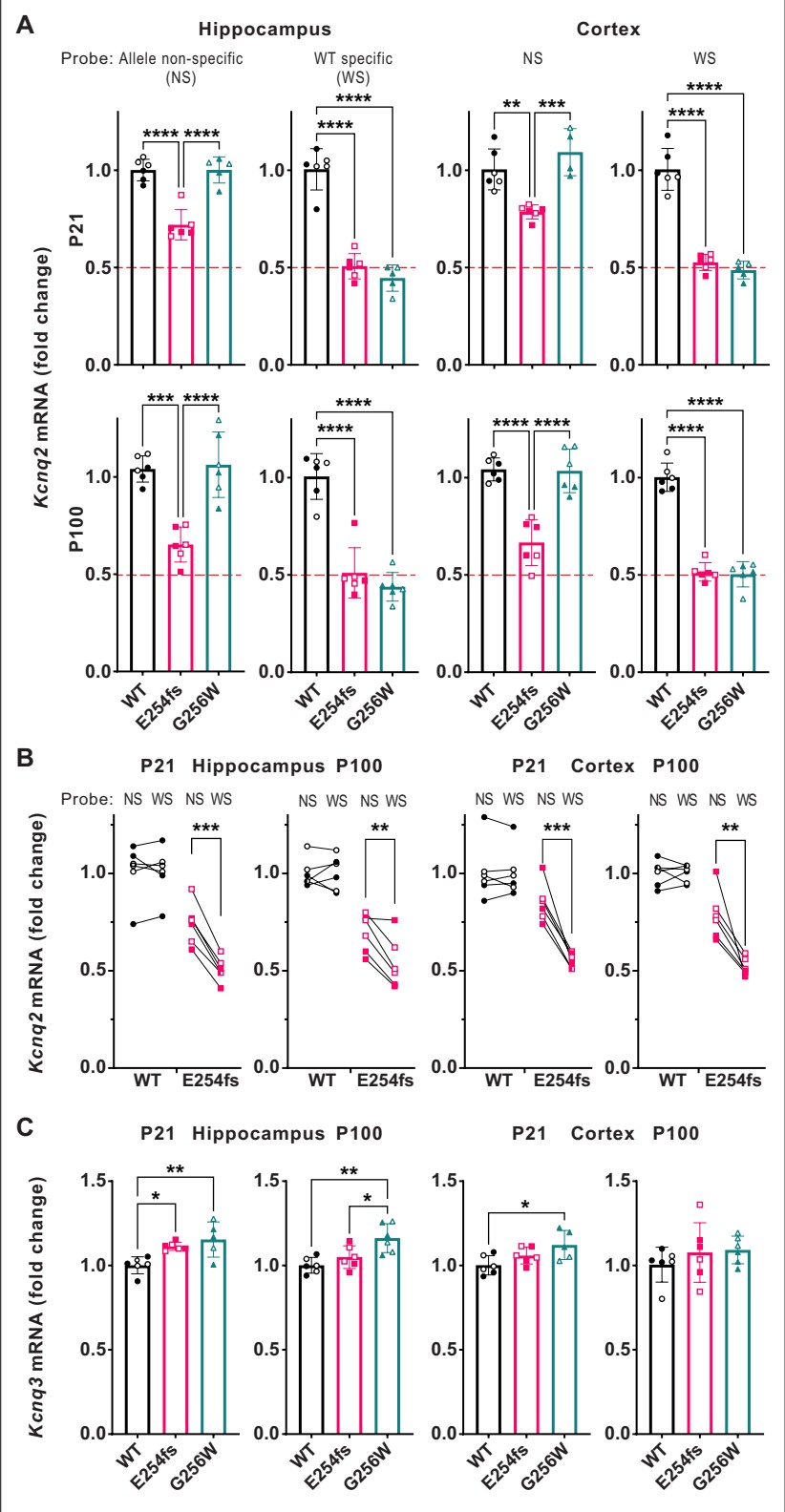

**Figure 7.** RT-qPCR shows incompletely efficient nonsense mediated decay of *Kcnq2* E254fs/+ mRNA and increased *Kcnq3* mRNA in E254fs/+ and G256W/+ mice. (**A**) Upper, *Kcnq2* mRNA levels in P21 hippocampus and neocortex. Levels were analyzed using TaqMan probes binding to both WT and variant *Kcnq2* alleles (allele non-selective, NS), or binding only to WT only (WT selective, WS). Total *Kcnq2* mRNA in E254fs/+ samples were

*Figure 7 continued on next page*

*Figure 7 continued*

significantly higher than 50% of WT (hippocampus: p=1.28 × 10⁻⁵, neocortex: p=3.5 × 10⁻⁷, one sample t-test). Lower, *Kcnq2* mRNA levels in P100 hippocampus and neocortex, using the probes as above. Total *Kcnq2* mRNA levels in E254fs/+ samples were significantly higher than the expected 50% of WT in hippocampus (p=0.0003) and neocortex: (p=0.0007). (**B**) Total and WT *Kcnq2* mRNA, tested in parallel, by individual. Age, sex (male, open symbols), brain region, genotype, and probe are indicated. In all four tissues tested, E254fs/+ mice have greater total *Kcnq2* mRNA than WT *Kcnq2* mRNA (P21 hippocampus, p=0.0001; P100 hippocampus, p=0.005; P21 neocortex, p=0.0007; P100 neocortex, p=0.0053; pairwise t-test). (**C**) In P21 G256W/+ mice, *Kcnq3* mRNA was significantly increased: 1.15-fold (+/-0.10, p=0.0043) in the hippocampus and 1.12-fold (+/-0.09, p=0.00213) in the neocortex. In P100 E254fs/+ mice, *Kcnq3* mRNA significantly increased (1.11+/-0.02 fold, p=0.0245) in the hippocampus only. One way ANOVA, *=p < 0.05, **=p < 0.005, ***=p < 0.0005. (See Supplemental Data for statistical test calculations).

The online version of this article includes the following source data and figure supplement(s) for figure 7:

**Source data 1.** The content of this file is used for the allele non-specific P21 *Kcnq2* data seen in **Figure 7A** and P21 *Kcnq3* data seen in **Figure 7C**.

**Source data 2.** The content of this file is used for the allele non-specific P100 *Kcnq2* data seen in **Figure 7A** *Kcnq3* data seen in **Figure 7C**.

**Source data 3.** The content of this file is used for the WT specific P21 *Kcnq2* data seen in **Figure 7A**.

**Source data 4.** The content of this file is used for the WT specific P100 *Kcnq2* data seen in **Figure 7A**.

**Source data 5.** The content of this file is used for **Figure 7B**.

**Figure supplement 1.** Measuring allele expression with allele-specific PCR amplification and cDNA Sanger sequencing.

---

The sum of all KCNQ2 band intensities showed an approximate 50% loss, like the monomer. This loss was expected for E254fs/+ mice, based on prior qPCR and absence of a band corresponding to the truncated protein. The loss was unexpected for G256W/+, although consistent with the hypothesis that mislocalized KCNQ2 subunits might have shortened half-lives in vivo. Yet, KCNQ3 protein levels in G256W/+ and WT mice were unchanged, despite the similar KCNQ2 and KCNQ3 redistribution from hippocampal pyramidal cell axons to somata exhibited immunohistochemically.

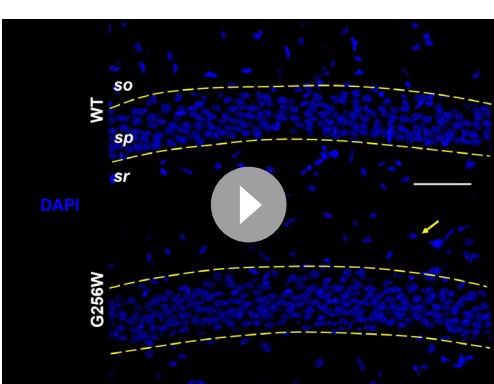

**Video 1.** Heterozygous G256W mice show reduced KCNQ2 and KCNQ3 labeling of CA1 pyramidal cell AISs and increased labeling of neuronal somata. Identically processed age P21 tissue sections of WT (upper) and G256W/+ (lower) mice; area CA1B was imaged under identical settings. Confocal image stacks are shown as maximal intensity projections. In the animation, channels for the indicated markers are allowed to fade into the next, enabling evaluation of colabeling. DAPI marks cell nuclei. AnkG strongly marks AISs and lightly labels somata and proximal apical dendrites. An arrow highlights one stratum oriens interneuron somatically labeled for KCNQ2 only. Labels: DAPI, 4',6-diamidine-2'-phenylindole; so, stratum oriens; sp, stratum pyramidale, sr, stratum radiatum. Scale: 50 µm.
https://elifesciences.org/articles/91204/figures#video1

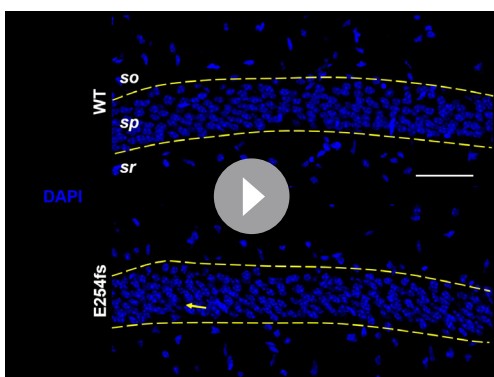

**Video 2.** In CA1, the KCNQ2 and KCNQ3 cellular and subcellular immunolabeling patterns appear similar for WT and heterozygous E254fs mice. Ankyrin-G marks position of AISs. KCNQ2 and KCNQ3 strongly label CA1 AISs in E254fs/+ mice, and do not show increased somatic labeling compared to WT. Highlighted by an arrow is one interneuron in stratum pyramidale that was somatically labeled for KCNQ2 only. Scale: 50 µm.
https://elifesciences.org/articles/91204/figures#video2

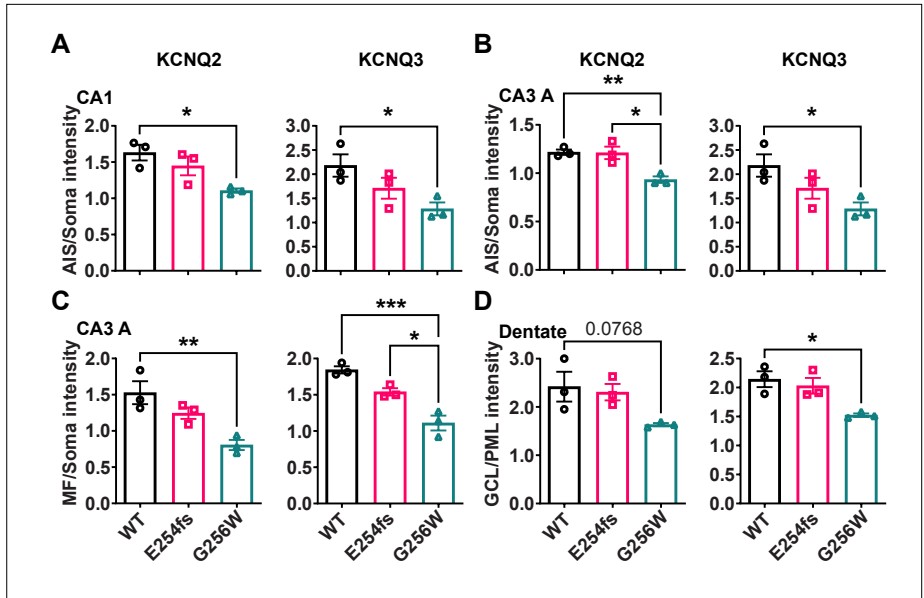

**Figure 8.** The ratios of axonal to somatic KCNQ2 and KCNQ3 labeling are reduced in CA1 and CA3 in heterozygous G256W mice. (**A, B**) The ratios of AIS to somatic immunofluorescence intensity is significantly reduced for KCNQ2 and KCNQ3 in CA1 (**A**) and CA3 (**B**) for G256W/+ but not E254fs/+ mice. (**C**) The ratio of mossy fiber to somatic KCNQ2 and KCNQ3 immunofluorescence intensity is reduced in the CA3 for G256W/+ but not E254fs/+ mice. (**D**) In the dentate gyrus, the ratio between GCL and PML intensity is significantly reduced for KCNQ3 but not KCNQ2 in G256W/+ but not E254fs/+ mice. n=3 per genotype. One way ANOVA, *=p < 0.05, **=p < 0.005, ***=p < 0.0005.

The online version of this article includes the following source data and figure supplement(s) for figure 8:

**Source data 1.** This file contains ROI data and calculations used for *Figure 8*.

**Figure supplement 1.** G256W/+ mouse interneurons in CA1 show somatic KCNQ2 labeling.

**Figure supplement 1—source data 1.** This file contains interneuron labeling data cited in results related to *Figure 8—figure supplement 1*.

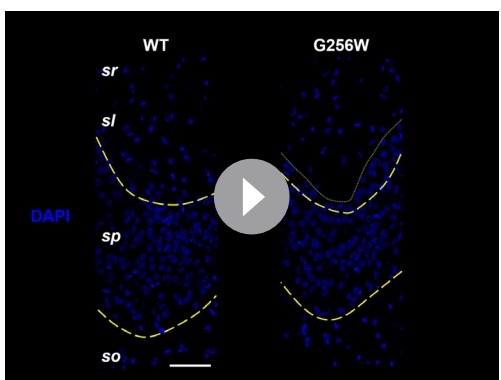

**Video 3.** Heterozygous G256W mice show increased CA3 pyramidal cell somatic labeling and reduced mossy fiber labeling for KCNQ2 and KCNQ3. Yellow lines demarcate the borders of sp; the sp-sl border is cut obliquely through the tissue section in the G256W/+ sample. PanNav strongly labels the unmyelinated axons of the mossy fibers in stratum lucidum of both samples. PanNav also labels the obliquely cut AISs of pyramidal cell neurons, which are mostly located within sp. Scale: 50 µm.
https://elifesciences.org/articles/91204/figures#video3

## Discussion

Learning how an ion channel variant alters human development is challenging and of interest to diverse stakeholders. The studies summarized here bring together clinical, protein structural, in vitro, and in vivo approaches. By carefully considering individual 1's medical history and then finding three additional affected individuals, we specified phenotype features to be modeled and generated independent clinical evidence of pathogenicity. Our lab studies highlighted three distinct mechanisms contributing to pathogenicity. First, we obtained evidence that Gly256 is part of an arch-shaped extracellular non-covalent bonding network between S5, the turret and the selectivity filter, a divergent and conserved structural feature of neuronal KCNQ2 and closely related subunits. This provided a novel hypothesis why a residue so distant from the pore might be pathogenic. Second, in heterologous cells, we found that expressing G256W with WT KCNQ2 and KCNQ3 subunits reduced currents consistent

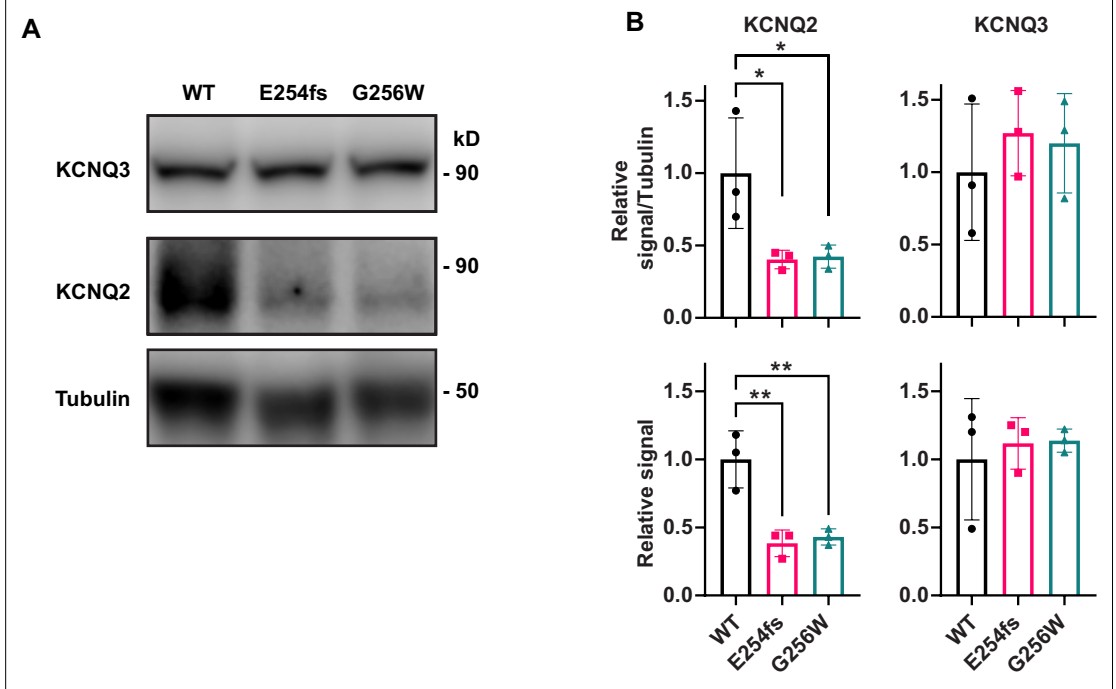

**Figure 9.** KCNQ2 protein is reduced in neocortex of P21 heterozygous E254fs and G256W mice. (**A**) Representative western blots for all three genotypes probed for tubulin, KCNQ2 and KCNQ3. (**B**) Quantified KCNQ2 monomer and KCNQ3 signal relative to WT, normalized to tubulin and by protein loaded as assayed by BCA. n=3 per genotype, all males. One way ANOVA, *=p < 0.05, **=p < 0.005.

The online version of this article includes the following source data and figure supplement(s) for figure 9:

**Source data 1.** This file has the quantification of signal intensities that appear in *Figure 9B*, *Figure 9—figure supplement 1C*.

**Source data 2.** This file is a zip folder containing uncropped images of the blots with relevant bands clearly labelled, the source files for *Figure 9A* and *Figure 9—figure supplement 1A, B*.

**Source data 3.** This zip folder contains raw blot image files (requires Image Studio Lite software) and unlabeled, intensity-windowed 8-bit tiff files used for blot figures.

**Figure supplement 1.** KCNQ2 antibodies, unlike KCNQ3, show complex electrophoretic banding pattern, and reduced levels in E254fs/+ and G256W/+ mice.

with the dominant-negative effects, though to an extent intermediate between previously described SLFNE and severe DEE variants. Third, we analyzed G256W/+ mice, learning that KCNQ2 and KCNQ3 proteins were quantitatively redistributed from hippocampal AIS and mossy fiber regions to somatic regions. KCNQ2 protein levels were reduced by about 50%, and CA1 neurons showed hyperexcitability. We performed experiments on E254fs/+ and G256W/+ mice in parallel. The contrasting results provide strong evidence that, in contrast to haploinsufficiency-linked SLFNE, in vivo dominant-negative effects of expressed G256W subunits drive DEE severity.

## Salient features of human G256W/+ related illness

Our clinical data, including de novo occurrence in three unrelated individuals and the informative phenotypic pattern observed in the European family including transmission from a mildly-affected mosaic parent (*Weckhuysen et al., 2012*; *Myers et al., 2018*), provide strong clinical evidence that KCNQ2 heterozygous G256W is pathogenic for DEE. Individual 1's history includes features characteristic of KCNQ2 DEE as previously described but include some issues meriting comment. The patient's epilepsy had an explosive early neonatal onset, with high seizure frequency and pharmacoresistance in the first 4 weeks of life. The potential that seizures began in utero was raised by the parents but was difficult to verify. Limited prior evidence suggests that, in some individuals born prematurely, onset of *KCNQ2*-related seizures is delayed until a post-conceptual age near term (*Ronen et al., 1993*). We note the issue, as in a previous report (*Berg et al., 2021*), to motivate broader, controlled survey. Although EEG records of individual 1 showed neither the invariant burst-suppression diagnostic of Ohtahara

syndrome or hypsarrhythmia, ACTH was given and remission followed shortly after. Because seizure remission in infancy is characteristic of *KCNQ2* DEE, the contribution made by ACTH is unclear and, among approved drugs, early trial of carbamazepine or oxcarbazepine is better supported (*Pisano et al., 2015*). Individual 1's neonatal seizures exhibited a range of onset patterns, arising unilaterally from either hemisphere or bilaterally. This is more varied than previously described (*Weckhuysen et al., 2013*; *Numis et al., 2014*). Postictal voltage attenuation was a feature of nearly all (92%) of unilateral and bilateral onset seizures; this is unusual for neonatal seizures and should be studied in a large sample, as it may be useful diagnostically. Despite achieving seizure remission by 1 month, an age earlier than remission in many individuals with SLFNE (*Ronen et al., 1993*; *Grinton et al., 2015*), individual 1's development showed global moderate to severe delay. This contrast between SLFNE and DEE outcomes despite impressive early seizure burden in both, seems to us to fit best within the emerging concept of 'developmental encephalopathy' (*Scheffer et al., 2017*; *Berg et al., 2021*), that is it is the persistent, strong KCNQ2 loss-of-function after seizure remission that drives developmental impairment, rather than seizure-induced injury.

## Expressed in heterologous cells, KCNQ2 G256W shows intermediate severity dominant-negative effects and ezogabine-responsiveness

How can these clinical features be explained by experiments, and what questions remain? We performed whole-cell patch recordings using two methods. One team performed automated planar patch recordings, another made manual pipette-based patch recordings. The teams were blinded to each other's work during data collection and analysis. Results showed good agreement, despite differences in sample preparation, cell selection, and recording protocols. Heteromeric KCNQ2/KCNQ3 channels showed no important differences between the methods. Because mean current density was larger using the manual patch method, likely due to selective study of high-expression cells in this method, the smaller KCNQ2 homomeric channel currents were studied by manual patch. Cells with 1:1 co-expression of WT KCNQ2 and G256W plasmids gave currents that were 30.1+/-0.1% of control. This is stronger conductance suppression than associated with KCNQ2 missense variants from SLFNE pedigrees (*Schroeder et al., 1998*; *Coppola et al., 2003*), but weaker suppression than exhibited by PGD de novo variants associated with very severe and profound impairment phenotypes (*Orhan et al., 2014*; *Tran et al., 2020*). Our manual-patch homomer recordings were insufficiently powered to achieve p<0.05 for null hypothesis rejection between the ~30% current observed and the 50% threshold for a 'dominant negative' effect. Because cell-to-cell variability in current density is expected in heterologous transient expression systems, future studies aiming to categorize a spectrum of variant and phenotypic subgroups by this parameter should be designed with this goal in mind.

Our CHO cell transient expression results indicate that inclusion of one G256W subunit per tetramer is insufficient to fully prevent channel activity. Under conditions mimicking G256W heterozygosity, ezogabine shifted voltage-dependence and produced fold increases in currents to an equal extent as in cells expressing WT KCNQ2 only. By contrast, currents under conditions mimicking G256W homozygosity (G256W+WT KCNQ3 co-expression) showed no augmentation by ezogabine. Although the basis for these differences between conditions deserve more study, the most clinically relevant condition is mimicking heterozygosity, and such currents were strongly rescued by acute ezogabine treatment. Our working model (*Figure 3A*) used simplifying assumptions: that heteromeric currents preferentially reflect activity of channels with 2 KCNQ2 and 2 KCNQ3 subunits, and that (overexpressed from viral promoters in CHO cells) KCNQ2 WT and G256W subunits are equally represented. Future studies using surface biotinylation and tagged and/or concatemerized subunit constructs can extend these results by testing assembly and surface trafficking directly in CHO cells, cultured neurons, and mice. These experiments have therapeutic implications, as highlighted by cystic fibrosis, where effective multidrug therapies combine agents promoting surface expression and with others enhancing open probability (*Middleton et al., 2019*).

## The utility of mouse *KCNQ2* DEE models, and some current limitations

We generated mice expressing G256W under the native mouse *Kcnq2* promoter. Homozygosity led to death at P0. G256W/+ mouse pups showed no differences in *Kcnq2* mRNA levels in hippocampus

or cortex at age P21, gained weight and acquired a set of behavioral skills equally with WT controls, but experienced recurrent generalized seizures. We gleaned several lessons.

In slices, CA1 pyramidal neurons exhibited normal resting membrane potentials but increased firing responses to current injections. These finding agree with prior studies of pyramidal cells of mice expressing other human pathogenic variants under the native *Kcnq2* promoter (*Singh et al., 2008*; *Biba-Maazou et al., 2022*). Both experimental data and modeling suggest that the combination of hyperexcitability and normal somatic biophysical properties may result from KCNQ2's low somato-dendritic and high AIS surface density (*Shah et al., 2008*; *Battefeld et al., 2014*; *Hu and Bean, 2018*). The 25–35% increases in CA1 firing rates we observed are similar to prior studies of both SLFNE and DEE variants (*Singh et al., 2008*; *Soh et al., 2014*; *Biba-Maazou et al., 2022*), suggesting to us that the important severity differences between those variant classes involve contributions distant from the soma, from other neuronal types, and/or dynamic network interactions. Indeed, G256W/+ mouse ictal EEGs showed striking similarities with those from human patients, including individual 1 (*Figure 1*), namely evolution through a phase of high voltage repetitive spiking to a long period of strong voltage attenuation. A similar transition to EEG voltage attenuation was observed in studies of mice with conditional homozygous *Kcnq2* deletion from *Emx1*-expressing (primarily, glutama-tergic) cells (*Aiba and Noebels, 2021*). Direct current EEG methods showed that ictal EEG attenua-tion in *Emx1-Kcnq2* null mice resulted from cortical spreading depolarization. Mice with deletion of Kv1.1/*Kcna1* had seizures without cortical spreading depression that could be converted to seizures with spreading depolarization by XE-991, the selective KCNQ family blocker. This highlights differ-ences between the Kv1 and Kv7/KCNQ channel families discussed further below. It is expected, but important to learn, that spontaneous seizures of mice with heterozygous DEE variants driven by the native *Kcnq2* promoter will show spreading depolarization. G256W/+ mouse seizures were some-times fatal (*Figure 6*, *Figure 6—video 1*), as found previously in mouse lines with strongly suppressed KCNQ2 current made via diverse genetic strategies (*Singh et al., 2008*; *Soh et al., 2014*; *Milh et al., 2020*; *Kim et al., 2021*). Recent surveys show no instances of SUDEP among human *KCNQ2* DEE cohorts, in contrast to Dravet syndrome, which has high SUDEP risk (*Berg et al., 2021*; *Donnan et al., 2023*). Our (JJ, SS, ECC) contacts with over 800 families linked to the KCNQ2 Cure Alliance and over 400 individuals enrolled in RIKEE have revealed no evidence of SUDEP. Study of *Kcnq2* mutant mice may illuminate human SUDEP mechanisms, nonetheless.

Parallel experiments performed on E254fs/+ and G256W/+ mice illuminated differences between SLFNE and DEE alleles that may contribute to differences in phenotype severity. In E254fs/+ mice, total *Kcnq2* mRNA was ~30% reduced from WT, but RT-qPCR experiments using a WT transcript-selective assay showed 50% loss. This result indicated that nonsense mediated decay was incom-pletely efficient for E254fs messages. Incomplete nonsense-mediated decay has been described previously, and may show variation between genes, variants, tissues, and cell types (*Zetoune et al., 2008*; *Dyle et al., 2020*). Although most *KCNQ2* frameshift alleles have been associated with SLFNE via clinical studies of pedigrees (*Millichap et al., 2016*; *Goto et al., 2019*), the potential for some premature termination variants to escape nonsense-mediated decay and exert dominant-negative protein effects is highlighted by our results. This should be considered when stop-gain variants present with DEE phenotypes, especially if de novo and in the c-terminal region. Both E254fs/+ and G256W/+ mice showed increases in *Kcnq3* mRNA. In *Kcnq2* haploinsufficiency, increased rela-tive numbers of KCNQ3 subunits may increase the proportion of KCNQ2 subunits forming KCNQ2/KCNQ3 heteromers. Such heteromers are activated at left-shifted voltages and have greater maximal open probability than KCNQ2 homomers. Increased incorporation of KCNQ3 subunits could compen-sate for reduced KCNQ2 subunit availability. Increased KCNQ3 expression will do less when half the KCNQ2 subunits are dominant-negative DEE variants that impair in vivo surface trafficking and ion conduction. G256W/+ showed a loss of KCNQ2 but not KCNQ3 brain protein in western blots, even though the subunits were similarly redistributed from hippocampal pyramidal cell AISs to somata. This difference is unexplained and intriguing.

The most prominent shared phenotype of human *KCNQ2* (or *KCNQ3*) loss-of-function variants is frequent recurrent neonatal seizures, usually beginning from the first 2–3 days of life (*Grinton et al., 2015*; *Millichap et al., 2016*), as experienced by individual 1. Our study, in agreement with a recent study of *Kcnq2*[T274M/+] knock in mice (*Milh et al., 2020*), leads us to conclude that as-yet-unknown differences between species render mice much less susceptible to neonatal seizures. Seizures are

highly penetrant in human SLFNE pedigrees, but no *Kcnq2* SLFNE mouse model known to date shows spontaneous seizures. G256W/+ and prior DEE model mice all show spontaneous generalized convulsive seizures and seizure-associated death (*Milh et al., 2020*; *Kim et al., 2021*). However, they do not exhibit the very explosive neonatal onset and transience found in children. This difference merits more investigation, including comparative interrogation of human postmortem brain mRNA and protein.

Impaired protein AIS localization resulting from KCNQ2 experimental and pathogenic missense variants in dissociated hippocampal cultures consisting predominantly of glutamatergic cells is long-known (*Chung et al., 2006*; *Pan et al., 2006*) but has not been previously observed under heterozygous conditions. KCNQ channels have been found in hippocampal parvalbumin and somatostatin interneurons though their roles remain very incompletely understood (*Cooper et al., 2001*; *Lawrence et al., 2006*; *Fidzinski et al., 2015*; *Soh et al., 2018*; *Jing et al., 2022*). Additional experiments using conditional *Kcnq2* alleles and interneuronal subtype-specific Cre's, and dedicated studies of G256W/+ mouse interneurons are warranted. Interneurons of WT mice can exhibit somatic KCNQ2 immunolabeling in aldehyde-fixed sections (*Cooper et al., 2001*). Therefore, our detection of potentially increased labeling in G256W/+ interneuron somata (*Figure 8—figure supplement 1*) is best viewed as a motivation for more studies. Our efforts to quantitate changes in KCNQ2 protein levels by western blot again showed the remarkably different biochemical features of native-tissue KCNQ2 and KCNQ3 subunits, previously observed in mouse and human brain (*Cooper et al., 2000*; *Kim et al., 2021*). The two subunits have very similar properties when expressed heterologously in CHO or HEK cells (*Tran et al., 2020*). Mechanisms for the strong detergent-resistance and complex SDS gel migration pattern of brain KCNQ2 are poorly understood.

## Towards better variant pathogenicity prediction

The rapid discovery of novel variants of uncertain clinical significance in *KCNQ2* and other Kv channel genes is a challenge for pathogenicity assessment. *KCNQ2* has been included in efforts to develop high-throughput channel variant prediction methods integrating multiple data types via artificial intelligence (*Boßelmann et al., 2022*; *Brünger et al., 2023*). Brunger et al. analyzed variants in a very diverse set of channels, finding that pathogenicity was significantly predicted by variant distance from the geometric centers of the ion pore and membrane. KCNQ2 G256 lies about 20 and 30 Å from these locations, respectively, and was therefore of lower predicted risk, yet it is pathogenic. We noticed that KCNQ2 G256 and its turret neighbors were both evolutionarily divergent (from the *Shaker*-like Kv channels, and from closer relative, KCNQ1) and strongly conserved (among amniote KCNQ2s). G256 is part of a co-evolving set of residues and bonding partners, implying shared functional importance. Because specific, long-distance functional coupling mechanisms are well-established for voltage-gated channels, for example between residues mediating voltage-sensing, gating, and selectivity (*Long et al., 2005*; *Hoshi and Armstrong, 2013*; *Zaydman et al., 2013*), more detailed structural information including consideration of bond-pairing, gating associated conformational change, conservation, and divergence may improve automated pathogenicity prediction accuracy.

## Conclusion: a model linking molecular mechanisms to pathogenic consequences

Individual 1's phenotype and variant location indicated that the pore turret could be important for KCNQ2 function in vivo. We tested this using structural modeling, heterologous cell voltage-clamp electrophysiology, and knock-in mice. Our results lead to a unifying model that is useful as it highlights a set of next experiments. The KCNQ2 selectivity filter could be stabilized by an extracellular hydrogen-bonded arch over the turret, and clinical heterozygous pathogenic variants could destabilize the arch and thereby, the pore. Prior studies of Kv channel slow and c-type inactivation highlight needed next experiments, including assessing the effects of $[K^+]_e$ on G256W current density, exploring the functional consequences of experimental mutations at other proposed turret network residues, and determining structures of mutant channels (*Hoshi and Armstrong, 2013*; *Reddi et al., 2022*; *Tan et al., 2022*; *Ye et al., 2022*; *Fernández-Mariño et al., 2023*; *Wu et al., 2023*). Ezogabine binds to a pocket on the pore domain between adjacent subunits, and its ability to correct structural consequence of PGD variants such as G256W can potentially be determined. The prominent EEG attenuation observed after KCNQ2 G256W human and mouse seizures, which is accompanied by spreading depolarization in *Emx1/Kcnq2* null mice (*Aiba and Noebels, 2021*), may be illuminating a distinctive

KCNQ2 cellular and network function, and its basis in a canonical molecular property. *Brown and Adams, 1980* made use of a surprising, inverted voltage-clamp protocol with a holding potential of –30 mV, and measured currents elicited during and after brief steps to more polarized potentials. This protocol isolated $I_M$ (primarily, KCNQ2/KCNQ3 current) because many other neuronal currents (notably including Kv1/*Sh* and Kv2 delayed rectifiers) have slow forms of voltage-dependent inactivation. Neurons that experience long-lasting depolarization after high activity or reversible injury must reboot through a temporally ordered process. Lack of voltage-dependent inactivation positions $I_M$ to make early contributions to such neuronal recovery and repolarization. By contrast, the contribution of NaV and Kv currents with slow voltage-dependent inactivation will be delayed until after repolarization. The growing set of construct-valid mice including G256W and T274M provide platforms for testing this model in diverse circuits and paradigms of impaired development beyond seizures.

## Materials and methods

### Human subjects

US and European patients were enrolled after parental consent in the RIKEE registry, a human subjects research protocol (H-34852) approved by the Institutional Review Board of Baylor College of Medicine. Individual 1 (USA) video-EEG review was performed under research protocol COMIRB 16–0152 approved by the Colorado Multiple Institutional Review Board, by a board-certified clinical neurophysiologist on recordings made as part of routine care. EEGs began after transfer from the birth hospital to a tertiary care center on the fifth day of life. The archived records included all seizures detected on initial clinical review of the continuous VEEG; interictal background was sampled in saved segments taken about every two hours. Individual 1 received ACTH, 60 units/m$^2$, twice daily for 2 weeks followed by a 4-week taper (*Takacs, 2023*). Individual 1 underwent clinical Sanger sequencing (GeneDx). Individual 2 was diagnosed using an in-house multigene panel and informatics workflow (case 14; *Jang et al., 2019*). Individuals 3 and 4 were diagnosed by a 32 gene NGS panel (Familial and Generalized Epilepsy Gene Panel, Center for Human Genetics Tübingen; CHGT). Mosaicism was diagnosed for Individual 4 at CHGT by lab-established procedures taking into account deviation from expected 50% read counts and the clinical history.

### Structural modeling interpretation

We examined structural models of KCNQ1, KCNQ2, KCNQ4 and KcsA using Mol*, UCSF Chimera and ChimeraX (*Goddard et al., 2018*). We visualized turret main chain bond angles in Pymol. We made pairwise turret alignments in Chimera and used Mol* to identify predicted noncovalent bonds in the turret region using default cut-off parameters (for hydrogen bonding: length 3.5, maximum angle deviation 45). We visualized model B-factor and surface electrostatic potential maps in ChimeraX, and assessed the local resolution and model congruence of the turret region of KCNQ2 in aligned electron density maps (EMD-30443, EMD-30446) using Chimera.

### Mouse video-EEG monitoring

Using methods previously described in detail, electrode implantation surgeries and EEG recordings were performed in-lab and at the Baylor College of Medicine IDDRC In Vivo Neurophysiology Core (*Creson et al., 2019*; *Jing et al., 2022*). A post-operative recovery period of 2 days was allowed before commencing video-EEG monitoring. During recordings, mice were able to explore their cages freely and had access to water and food. Seizures counts were made by video and EEG review under band pass filtering (1–59 Hz). Seizure events were clipped and saved.

### Generation and use of KCNQ2 heterozygous G256W and E254fs mice
#### Crispr and mouse husbandry

All animal procedures were performed in accordance with protocols reviewed and approved by the Institutional Animal Use and Care Committees of the Jackson Laboratories (99066), Baylor College of Medicine (AN602, AN-5531), and the University of Connecticut (A22-042). KCNQ2 G256 is conserved in all amniotes (*Figure 2C*), but the codons are not (C57BL/6J mouse: GGU, human: GGG). Therefore, we made two base substitutions to introduce W256 (*Figure 7—figure supplement 1D*). C57BL/6J single cell zygotes were microinjected with a 123-nt oligonucleotide donor sequence:

5'-tggtacattggcttcctctgcctcatcctggcctcatttctggtgtacttggcagaaaagTgGgagaatgaccactttgacacc
tacgcagatgcactctggtggggtctggtaagtcctggt-3'

containing two nucleotide differences (capitalized) to change the glycine GGT codon to a tryp-
tophan TGG codon (G256W). Co-injected with the donor oligonucleotide was the *Kcnq2* exon 5
targeting guide RNA; 5'-TCTGGTGTACTTGGCAGAAA-3' and Cas9. Incorporation of the G256W
mutation into the genome would change the AGG PAM recognition sequence to AGT and prevent
retargeting of the modified allele. Founders were generated after embryo transfer into pseudopreg-
nant C56BL/6 J females and screened by sequence analysis of the *Kcnq2* exon 5 genomic DNA using
PCR and sequencing primers 5'-GGGATTCCATCCTCCAAGTC-3' and 5'-CCAGCCCAGCCTAAAG
ACA-3'. Among 40 progeny that carried the desired G256W-encoding mutation, a single founder
female was identified f carrying in trans a frameshifting deletion c.760_767; p.Glu254fs*16 (deletion of
the AAAAGGG nucleotides overlapping with the PAM sequence, *Figure 4A*). Lines were established
from both alleles after three successive backcrosses to C57BL/6J mice and designated as Jax Stock
numbers 029407 and 029408, respectively. Genotype by sequencing protocols were later replaced
with RT-PCR fluorometric probe assays, specific for the WT, G256W and E254fs deletion mutations
(Transnetyx).

## Survival analysis

Only animals that were backcrossed five or more times were included in the survival analysis. During
the COVID pandemic, many animals were euthanized in response to mandated colony reduction and
research suspension; these animals were also excluded from analysis. Animals were censored if they
were euthanized due to other (non-seizure related) health reasons or to not exceed IACUC approved
animal usage numbers. All early deaths in G256W mutants appeared to potentially be the result of
fatal seizures, as all recovered carcasses exhibited a stereotyped posture with symmetrically flexed
forelimbs and extended hindlimbs.

## Developmental milestone assays and behavioral seizure recording

Developmental milestone assays were performed following previously described methods (*Hill
et al., 2008*). Pups were bred group-timed matings from first time breeders to avoid single litter
effects. To test surface righting, an investigator held an individual mouse pups gently on its back, then
measured the time required for the pup to flip onto its abdomen after being released. For the nega-
tive geotaxis test, an investigator placed the mouse on a 45° sloped surface with its head pointing
downhill, then measured the time required for the pup to turn its body 180°. To test cliff aversion, an
investigator positioned the mouse pup's forepaws on the edge of a smooth surface 4" above a table,
then recorded the time for the pup to turn its body and take a step away from the edge. The P10
seizure (*Figure 4D–H*, *Figure 4—video 1*) was observed during a pilot study of pup open field motor
behavior conducted as previously described (*Bass et al., 2020*).

## RT-qPCR

All P100 experiments included the same three males and three female mice per genotype. For
P21, three males and three females were collected for WT and E254fs/+ mice, while two males and
three females were collected for G256W/+ mice. Animals were deeply anesthetized with isoflurane
and decapitated. Brains were removed and neocortical and hippocampal regions dissected on ice.
Samples were frozen on dry ice and stored at –80 °C. RNA was isolated from brain tissue (RNeasy Lipid
Tissue kit, QIAGEN) and stored at –80 °C until use. RNA (1 μg/sample) was used to generate cDNA
(SuperScript III Reverse Transcriptase, Invitrogen), which served as template for qPCR using Thermo
Fisher Scientific TaqMan gene expression assays with the TaqMan Fast Advanced Master Mix. Assays
were run on the QuantStudio 3 PCR system (Applied Biosystems), using *Gapdh* as reference and the
ΔΔCt quantification method (*Livak and Schmittgen, 2001*). For measuring total *Kcnq2* and *Kcnq3*
mRNA, the PCR amplicon spanned exons 5 and 6 (*Kcnq2* assay) or exons 4 and 5 (*Kcnq3*); primers and
probes recognized sequences common to all known splice isoforms (*Pan et al., 2001*). For the qPCR
assay discriminating between *Kcnq2* WT and variant alleles, we developed a custom TaqMan assay
(Thermo Fisher Scientific) where the fluorescent probe binding site included all 8 WT bases deleted
or substituted in the mutant alleles. For *Figure 7*, each (n=303) individual data point is the mean of

three replicate wells. One additional result was determined post hoc to be an outlier by the Grubbs' test and is not shown (P21 male neocortex, *Kcnq2*[G256W/+], allele non-specific probe, *Figure 7A*; thus n=1 male, and n=3 females). Each of the 16 graphs summarizes experiments performed in parallel on a PCR plate. Thermo Fisher Scientific TaqMan Assay ID's are listed in Appendix 1—key resources table.

## cDNA Sanger sequencing

cDNA made from 1000 µg of total RNA was amplified using primers spanning *Kcnq2* exons 4 through 7 (Apex Hot Start 2 X Master Mix; Blue Apex Buffer 1, ID No. 5200600–1250). cDNA was amplified using the following touchdown protocol: 1. 95 °C 15 min 2. 94 °C 20 s 3. 64 °C 30 s (- 1 °C every cycle) 4. 72 °C 30 s 5. Go to step 2 (repeat 5 times) 6. 94 °C 20 s 7. 59 °C 30 s 8. 72 °C 30 s 9. Go to step 6 (repeat 34 times) 10. 72 °C 10 min 11. 4 °C hold. The PCR reaction product was cleaned (Zymo DNA Clean & Concentrator Cat. No D4013) and sequenced (Genewiz) using the same primers used for amplification.

# Heterologous expression and CHO cell recording

## Cell lines

Chinese hamster ovary (CHO) cells were used for heterologous expression of KCNQ channels due to their low background K$^+$ currents (*Gamper et al., 2005*). In manual patch recordings, we used CHO cells obtained as a gift from the laboratory of Dr. Mark Shapiro (*Gamper et al., 2005*) and maintained on Dulbecco's modified Eagle's medium (Gibco) supplemented with 10% fetal bovine serum (FBS, Gibco). As previously described (*Vanoye et al., 2022*), automated patch recording used the Flp-In-CHO cell line (Invitrogen; ThermoFisherScientific) maintained on F-12 medium/10% FBS (Atlanta Biologicals). Cultures were monitored daily by light microscopy for consistent cell morphology and culture growth rates, and for transfection efficiency by GFP expression using a co-transfected pcDNA3-GFP-expressing plasmid. CHO cells are not listed in the register of commonly misidentified cell lines maintained by the International Cell Line Authentication Committee (*Capes-Davis et al., 2010*).

## Automated patch

Methods for expression and recording in CHO cells were previously described in detail (*Vanoye et al., 2022*). Human KCNQ2 cDNA (GenBank accession number NM_172108) in pIRES2-EGFP (BD Biosciences-Clontech, Mountain View, CA, USA) was used as template for in vitro mutagenesis. A stable line expressing human KCNQ3 (GenBank accession number NM_004519) was generated as described and maintained under dual selection with Zeocin (100 µg/ml) and hygromycin B (600 µg/ml). Plasmids with WT KCNQ2 or G256W cDNA were introduced into the KCNQ3-expressing CHO cell stable line by electroporation (Maxcyte STX; MaxCyte Inc, Gaithersburg, MD, USA). Automated patch clamp recording was performed using the Syncropatch 768 PE platform using PatchController384 V.1.3.0 software (Nanion Technologies, Munich, Germany). Pulse protocols were performed before and after addition of ezogabine (10 µM, Sigma-Aldrich) and, subsequently, XE-991 (25 µM, Abcam, Cambridge, MA; or TOCRIS, Minneapolis, MN). Currents reported are XE-991-sensitive currents, calculated by subtraction.

## Manual patch

Mutagenesis, cell culture, transfection, and manual patch recordings were performed as previously described (*Tran et al., 2020*). CHO cells plated on cover slips were recorded at room temperature (20–22°C), 2–3 days post-transfection, using an Axopatch 200B amplifier (Molecular Devices), pCLAMP v.9, a cFlow perfusion controller and mPre8 manifold (Cell MicroControls), and glass micropipettes (VWR International) with 1–4 MΩ resistance. The extracellular solution consisted of (in mM): 138 NaCl, 5.4 KCl, 2 CaCl$_2$, 1 MgCl$_2$, 10 glucose, 10 HEPES, pH 7.4 with NaOH. Pipette solution contained (in mM): 140 KCl, 2 MgCl$_2$, 10 EGTA, 10 HEPES, and 5 Mg-ATP, pH 7.4 with KOH. Series resistance was compensated by 70% after compensation using Axopatch 200B fast and slow capacitance controls. Currents were digitally sampled at 10 kHz and filtered at 5 kHz using a low-pass Bessel filter. For voltage-activation experiments, cells were held at −80 mV and depolarized in 10 mV incremental steps from −80 to +40 mV for 1 s, then stepped to 0 mV for 60ms, followed by a 20 s, −80 mV

interpulse. Tail currents were fitted using the Boltzmann function in Prism to obtain the half-maximum activation voltage ($V1/2$) and the slope factor ($k$).

## Whole Cell CA1 pyramidal cell recording

### Acute brain slice preparation

For all electrophysiological experiments, we used P12–P15 mice of both sexes. The mice were anesthetized with isoflurane and quickly euthanized through decapitation. Subsequently, their brains were extracted and placed in a chilled cutting solution composed of 26 mM $NaHCO_3$, 1.25 mM $NaH_2PO_4$, 2.5 mM KCl, 0.5 mM $CaCl_2$, 7 mM $MgCl_2$, 10 mM dextrose, and 210 mM sucrose. To record from the hippocampus, 300 µm slices were cut horizontally using a microtome (Leica VT1200S). These slices were then moved to a holding chamber containing artificial cerebrospinal fluid (ACSF), which contained 125 mM NaCl, 2.5 mM KCl, 1.3 mM $MgCl_2$, 1 mM $NaH_2PO_4$, 26 mM $NaHCO_3$, and 12 mM dextrose, 1.5 mM $CaCl_2$ was supplemented the day of the recording. Both the cutting solution and ACSF were consistently saturated with a mixture of 95% $O_2$ and 5% $CO_2$. The brain slices in ACSF underwent a 30 min incubation in a 37 °C water bath, followed by at least one hour at room temperature prior to the actual recordings. Subsequently, the slices were transferred to a recording chamber where the temperature was maintained at 30–32°C using a Warner Instruments TC 324 C temperature controller. Continuous perfusion of ACSF into the chamber was achieved through a peristaltic pump.

### Electrophysiological recordings

We used borosilicate glass electrodes with resistances of 2–4 MΩ for conducting whole-cell recordings. Current clamp recordings were done on ventral CA1 pyramidal neurons of the hippocampus. Recording pipettes were filled with an internal solution composed of 130 mM $CH_3KO_4S$, 10 mM KCl, 4 mM NaCl, 4 mM Tris-phosphocreatine, 10 mM HEPES, 4 mM Mg-ATP, and 0.4 mM Na-ATP. These neuronal recordings took place in the presence of synaptic blockers, including 100 µM picrotoxin to inhibit GABAA receptor-mediated inhibitory responses, 4 µM NBQX to block AMPA-mediated responses, and 10 µM D-AP5 to inhibit NMDA-mediated responses. All current clamp recordings were performed using a Multiclamp 700B amplifier (Molecular Devices) with bridge balancing engaged to compensate for input resistance. Cells with an access resistance of less than 20 MΩ were selected for both recording and subsequent analysis. To evaluate intrinsic excitability and action potential waveform characteristics, a depolarizing current injection ranging from +25 pA to +325 pA was applied in +25 pA increments, each lasting for 1 s with 15 s intervals between sweeps. To determine the cell's input resistance, a series of hyperpolarizing steps spanning from –100 pA to 0 pA were administered in –25 pA increments, with each hyperpolarizing step maintained for 1 s and no intervals between sweeps. The holding membrane potential was set at –65 mV prior to the step protocols by injecting small DC current through the pipette. Resting membrane potential was measured at I=0 soon after breaking into the cell. AP properties were determined by the 1st action potential in a step protocol. Data were sampled at 50 kHz, with the Bessel filter set at 10 kHz. Data acquisition for all electrophysiology experiments was executed using a Digidata 1440 A system and pClamp software (versions 10.2–11.2; RRID:SCR_011323).

## Immunohistochemistry

We trialed three alternative tissue preparation protocols: unfixed cryosections with or without ice-cold methanol pre-staining steps (*Devaux et al., 2004*; *Pan et al., 2006*), and a previously used protocol using weak formaldehyde perfusion fixation followed by microwave/citrate antigen retrieval (*Shah et al., 2008*). The formaldehyde/antigen retrieval method more reliably yielded intact tissue on slides, an outcome that was important for the analysis we planned comparing multiple matched sections per animal and multiple animals per each of 3 genotypes. Accordingly, we adopted that approach for experiments used for quantitation shown here. After establishment of deep anesthesia (300 mg/kg ketamine/30 mg/kg xylazine IP), mice were transcardially perfused with 20 mL of sterile ice-cold PBS, followed by 20 mL of ice cold 2% formaldehyde in PBS, freshly prepared from paraformaldehyde 20% aqueous solution (Electron Microscopy Sciences). The brain was removed from the skull, post-fixed on ice for 60 min, embedded (OTC Tissue Tek Compound, Sakura Finetek), and stored at 80 °C. Coronal 40 µm sections including dorsal hippocampus were cut on a cryostat, transferred to slides (Fisher SuperFrost Plus, Fisher Scientific), and stored at –80 °C. Slides bearing brain sections at

matched rostrocaudal positions of the dorsal hippocampus were thawed for 10 min at room temperature (RT), subjected to citrate/microwave antigen retrieval, washed with PBS for 10 min, and blocked (PBS, 5% normal goat serum, 0.5% Triton-X 100). Sections were incubated overnight with blocking buffer containing primary antibodies affinity-purified rabbit anti-KCNQ2 N-terminal antibody, 1:200, KCNQ3 N-terminal antibody, 1:500, and either mouse anti-AnkG IgG2a, clone N106/36, Neuromab/Antibodies, Inc, 75–146, 1:1000, or mouse anti-PanNav IgG1 (hybridoma culture supernatant, 1:200, gift of Jim Trimmer, commercially available as affinity-purified IgG, Sigma K58/35 S8809) using incubation chambers formed with CoverWell Gaskets (Thermo Fisher C18150) to prevent drying. Wash steps used PBS with 0.05% Triton-X100. Slides were coverslipped using ProLong Gold with DAPI (Thermo Fisher P36931).

## Quantification of KCNQ2 and KCNQ3 immunolabeling intensity

Confocal imaging was performed on trios of slides differing in genotype (WT, G256W/+ and E254fs/+) and immunostained in parallel for KCNQ2. KCNQ3, and PanNaV. The slides were labeled only by numerical codes--the experimenter performing the imaging and quantification analysis was blinded to genotype throughout. Sample trios were imaged under the exact same settings. Image stacks for analysis were collected in CA1 B, CA3A, and the dentate. Prior to final image collection, nearby optical fields were used to adjust acquisition parameters to optimize dynamic range and eliminate detector saturation for any sample within the set. Image stacks (pixel XY dimensions, 0.31 μm, 1024x1024 pixels/image, 21 images/stack, z-step of 0.500 μm) were generated using a Nikon C2 confocal microscope (20x0.75 NA Nikon planapo lens, and 2 X zoom), running NIS Elements 4.0 AR. Quantitation was performed on maximal projections of the confocal stacks. Regions of interest (ROIs) were marked by DAPI (for somata), and PanNaV (for AISs and mossy fibers in the dentate hilus and CA3). In CA1 and CA3, the somatic ROIs were portions of stratum pyramidale without AISs. In the dentate, the fine AISs could not be excluded as they lie within the granule cell layer (*Shah et al., 2008*), but compared to CA1, they are very thin and inconspicuous under 20 x magnification. KCNQ2 and KCNQ3 color channels were masked during manual marking of somatic and AIS or mossy fiber ROIs. The ROI-setting procedure used two steps. First, the blinded investigator used the freehand drawing tool, and PanNaV to mark the AIS- and mossy fiber containing ROIs, and used DAPI labeling to delimit the remaining portions of the pyramidal cell or granule cell layers excluding zones of overlap. Second, within these regions, 5 smaller, equally sized, rectangular ROIs were placed, avoiding any visible AISs in the soma-containing ROIs. The intensity values for KCNQ2 or KCNQ3 were exported and averaged across all ROIs for each sample. AIS or MF mean ROI intensities were then divided by those of adjoining somata regions (e.g., the average ROI intensity for KCNQ2 or KCNQ3 in the AIS region of CA1 divided by the average intensity in the CA1 pyramidal cell layer from the same image stack). This analysis was performed on two image stacks per animal per region (one per hemisphere), and 3 animals per genotype. Male and female mice were both included since completed qPCR and preliminary immunohistochemical experiments showed no sex differences. Putative CA1 interneurons were identified by size, location, somatic labeling for AnkG and/or KCNQ2, and presence of a neighboring AIS.

## Immunoblotting

For experiments shown in *Figure 9* and *Figure 9—figure supplement 1*, tissues from individual animals were processed separately, in parallel, to allow for comparison of biological replicates of each genotype. Animals were deeply anesthetized with isoflurane and decapitated. Brains were removed and neocortical and hippocampal regions dissected on ice. Samples were frozen on dry ice and stored at –80 °C. Cortex samples were weighed and homogenized with a glass-glass homogenizer in 10 volumes (w/v) of RIPA (150 mM NaCl, 1.0% Triton X-100, 0.5% sodium deoxycholate, 0.1% SDS, 50 mM Tris HCl pH 7.4, Pierce EDTA-free protease inhibitor). Homogenates were sonicated (Branson model 450, Cat. No. 15338553) with a 1/8" microtip (Cat. No. 101063212) at 40% amplitude for 10 sec three times, aliquoted and stored at –80 °C. Protein concentration was determined by bicinchoninic acid assay (BCA, Pierce). Homogenate aliquots were thawed on ice, supplemented with SDS sample buffer (Licor) and dithiothreitol (10 mM final concentration), incubated at 37 °C for 30 min, mixed, and loaded on 7.5% or 4–15% Mini-Protean TGX gels (Biorad). Resolved proteins were electro-transferred to PVDF. We incubated filters in 5% non-fat dry milk (Carnation) in Tris-buffered saline with 0.2% Tween (TBST) for one hour, then overnight in the same buffer with

affinity-purified anti-KCNQ2 or anti-KCNQ3 primary antibodies (*Cooper et al., 2001*; *Jin et al., 2009*) at 1:400 and 1:1000, respectively. After washing, filters were incubated with HRP conjugated secondary antibodies. Blots were visualized by enhanced chemiluminescence (Amersham Cytiva ECL Prime), and imaged using a CCD-based gel documentation system (LiCor Odyssey XF). Although blots were probed in parallel for tubulin as a loading control (Sigma T6199) protein loading required for channel subunit detection was outside the tubulin linear range (*Kirshner and Gibbs, 2018*). Quantification was performed using normalization to tubulin and based on equal sample loading (i.e., to the BCA) with similar results. Quantification of band intensities was performed using Image Studio (LiCor, v.5.2). Background was subtracted by the Image Studio software's individual band 4-sided surround method.

## Statistical analysis

### Development, qPCR, Immunohistochemistry, and western blot

Statistical tests were applied using Prism version 9.5.1 (GraphPad Software, San Diego, California USA). Data distribution normality was analyzed using the D'Agostino-Pearson test. In E254fs/+ mice, for comparison of *Kcnq2* mRNA levels versus the expected 50%, a one-sample t-test was conducted. For other RT-qPCR, western blot, and immunohistochemistry (AIS and mossy fiber vs somatic intensity) comparisons, group differences were tested by two-way ANOVA to determine if genotype, sex, and/or the interaction between the two had a significant effect. As sex was found not to significantly affect the data, it was dropped as a factor and a one-way ANOVA was performed with genotype as the only factor. For mouse developmental milestone data, the same approach was used except for using a repeated measures ANOVA. Tukey's HSD post hoc test was used to assess pairwise significance between genotypes.

### Electrophysiology

Statistical tests were applied using Prism version 9.5.1 (GraphPad Software, San Diego, California USA). Data distribution normality was analyzed using the D'Agostino-Pearson test. For in vitro patch clamp recordings, I/V and G/V curves were analyzed using a two-way repeated measures ANOVA with the Geissner-Greenhouse correction, matched values from each recorded cell were stacked into a subcolumn. Test potential (voltage) and expression (e.g., Q2 WT + G256W) were defined as factors. Pairwise comparisons were made between expression groups at each test potential, with Tukey's post hoc analysis correcting for multiple comparisons. For comparing heterozygous homomeric currents to 50% of WT, a t-test was performed between the average current densities at +40 mV. For slice recording data, the same approach was used but with action potential number and current injection being defined as factors.

## Acknowledgements

This work was supported by an American Epilepsy Society Predoctoral Fellowship funded in part by Wishes for Elliott (TJA); by Citizens United for Epilepsy Research, the Jack Pribaz Foundation, the KCNQ2 Cure Alliance, the Miles Family Fund, and R01 NS49119 (ECC), by the NINDS Epilepsy Center without Walls U54 NS108874 (CV, ALG, NV, AT, KS, ECC); by NIH Grants R01 NS101596 and NS108874 (to AVT); by NS096029 and MH126953 (to AM); by NINDS U54 OD020351 (AZ, CL); by R01 NS29709 (JLN); and by K08 NS110924 (VK). The Jackson Laboratory scientific services are supported in part through the National Cancer Institute's Cancer Core Grant P30 CA034196. We acknowledge contributions from The Jackson Laboratory Genome Engineering Team and the Transgenic Geno-typing Services for assistance and consultation during the generation of the *Kcnq2* mutant mice. The BCM Mass Spectrometry Proteomics Core was supported by the Dan L Duncan Comprehensive Cancer Center NIH Award P30 CA125123, CPRIT Core Facility Award RP210227, Intellectual Development Disabilities Research Center P50 HD103555, and an NIH High End Instrument Award S10 OD026804. EEG at the Baylor College of Medicine Intellectual and Developmental Disabilities Research Center Neuroconnectivity Core was supported by NICHD P50 HD103555. The BCM CryoEM Core and ZW were supported by NIH R01 GM143380 and R01 HL162842, the Welch Foundation Q-2173–20230405, and by a CPRIT Core Facility Award RP190602.

# Additional information

## Competing interests

Alfred L George: receives funding from Biohaven Pharmaceuticals and is on the Scientific Advisory Board of Tevard Biosciences. Edward C Cooper: served as a consultant to Xenon Pharmaceuticals and Knopp Biosciences. This activity has been reviewed and approved by Baylor College of Medicine according to its Conflict of Interest policies. The other authors declare that no competing interests exist.

## Funding

| Funder | Grant reference number | Author |
| --- | --- | --- |
| American Epilepsy Society | Predoctoral Fellowship | Timothy J Abreo |
| Citizens United for Research in Epilepsy | Pediatric Epilepsy Award | Edward C Cooper |
| Jack Pribaz Foundation | | Edward C Cooper |
| National Institute of Neurological Disorders and Stroke | R01 NS49119 | Edward C Cooper |
| National Institute of Neurological Disorders and Stroke | R01 NS101596 | Heun Soh<br>Nissi Varghese<br>Kristen Springer<br>Anastasios Tzingounis |
| National Institute of Neurological Disorders and Stroke | R01 NS108874 | Heun Soh<br>Nissi Varghese<br>Kristen Springer<br>Anastasios Tzingounis |
| National Institute of Neurological Disorders and Stroke | K08 NS096029 | Atul Maheshwari |
| National Institute of Mental Health | MH126953 | Atul Maheshwari |
| National Institute of Neurological Disorders and Stroke | U54 OD020351 | Aamir R Zuberi<br>Cathleen M Lutz |
| National Institute of Neurological Disorders and Stroke | R01 NS29709 | Jeffrey L Noebels |
| National Institute of Neurological Disorders and Stroke | U54 NS108874 | Heun Soh<br>Nissi Varghese<br>Carlos G Vanoye<br>Kristen Springer<br>Alfred L George<br>Anastasios Tzingounis<br>Edward C Cooper |
| National Cancer Institute | P30 CA034196 | Aamir R Zuberi<br>Cathleen M Lutz |
| Eunice Kennedy Shriver National Institute of Child Health and Human Development | P50 HD103555 | Edward C Cooper |
| National Institute of General Medical Sciences | R01 GM143380 | Zhao Wang |
| National Heart, Lung, and Blood Institute | R01 HL162842 | Zhao Wang |
| Welch Foundation | Q-2173-20230405 | Zhao Wang |

| Funder | Grant reference number | Author |
| --- | --- | --- |
| Cancer Prevention and Research Institute of Texas | RP190602 | Zhao Wang |
| KCNQ2 Cure Alliance | | Edward C Cooper |
| Miles Family Fund | | Edward C Cooper |
| National Institute of Neurological Disorders and Stroke | K08 NS110924 | Vaishnav Krishnan |

The funders had no role in study design, data collection and interpretation, or the decision to submit the work for publication.

## Author contributions

Timothy J Abreo, Conceptualization, Data curation, Formal analysis, Investigation, Methodology, Writing – original draft, Writing – review and editing; Emma C Thompson, Anuraag Madabushi, Heun Soh, Ana G Chavez, Data curation, Formal analysis, Investigation, Methodology, Writing – review and editing; Kristen L Park, Edward C Cooper, Conceptualization, Resources, Data curation, Formal analysis, Supervision, Funding acquisition, Investigation, Methodology, Writing – original draft, Project administration, Writing – review and editing; Nissi Varghese, Carlos G Vanoye, Data curation, Formal analysis, Investigation, Methodology, Project administration, Writing – review and editing; Kristen Springer, Jim Johnson, Conceptualization, Resources, Data curation, Formal analysis, Funding acquisition, Investigation, Writing – original draft, Writing – review and editing; Scotty Sims, Zhigang Ji, Conceptualization, Resources, Data curation, Formal analysis, Funding acquisition, Investigation, Methodology, Writing – review and editing; Miranda J Jankovic, Data curation, Investigation, Methodology, Writing – review and editing; Bereket Habte, Aamir R Zuberi, Resources, Data curation, Investigation, Methodology, Writing – review and editing; Cathleen M Lutz, Conceptualization, Resources, Data curation, Investigation, Methodology, Writing – original draft, Writing – review and editing; Zhao Wang, Vaishnav Krishnan, Conceptualization, Resources, Data curation, Supervision, Investigation, Methodology, Writing – original draft, Writing – review and editing; Lisa Dudler, Stephanie Einsele-Scholz, Resources, Supervision, Investigation, Methodology, Writing – review and editing; Jeffrey L Noebels, Alfred L George, Resources, Data curation, Formal analysis, Supervision, Funding acquisition, Investigation, Methodology, Writing – review and editing; Atul Maheshwari, Anastasios Tzingounis, Conceptualization, Resources, Data curation, Formal analysis, Supervision, Funding acquisition, Investigation, Methodology, Project administration, Writing – review and editing

## Author ORCIDs

Timothy J Abreo ⓘ https://orcid.org/0000-0001-5675-7932
Kristen L Park ⓘ https://orcid.org/0000-0001-5527-3047
Carlos G Vanoye ⓘ https://orcid.org/0000-0002-4935-1122
Aamir R Zuberi ⓘ https://orcid.org/0000-0003-0252-9760
Alfred L George ⓘ https://orcid.org/0000-0002-3993-966X
Atul Maheshwari ⓘ https://orcid.org/0000-0003-3045-7901
Edward C Cooper ⓘ https://orcid.org/0000-0003-3672-8442

## Ethics

Human subjects were enrolled after parental informed consent in the RIKEE registry, a human subjects research protocol H-34852 which includes consent to publish, approved by the Institutional Review Board of Baylor College of Medicine. Review of human EEG data was performed under protocol COMIRB 16-0152 approved by the Colorado Multiple Institutional Review Board.

This study was performed in strict accordance with the recommendations in the Guide for the Care and Use of Laboratory Animals of the National Institutes of Health. All animal procedures were performed in accordance with protocols reviewed and approved by the Institutional Animal Use and Care Committees of the Jackson Laboratories (99066), Baylor College of Medicine (AN602, AN-5531), and the University of Connecticut (A22-042). All surgery was performed under isoflurane anesthesia, and every effort was made to minimize suffering.

Reviewer #1 (Public Review): https://doi.org/10.7554/eLife.91204.3.sa1
Reviewer #3 (Public Review): https://doi.org/10.7554/eLife.91204.3.sa2
Author response https://doi.org/10.7554/eLife.91204.3.sa3

## Additional files

### Supplementary files

Source data 1. Single marker greyscale images and selected 2 color merged images of CA1. In upper panels, the simultaneously acquired AnkG color channel is not included in the merge, for clarity. In middle and lower panels, the yellow boxed regions of upper panels are shown as single channel greyscale and as the indicated merges of 2 channels. In lower merge images, yellow arrows indicate KCNQ2/KCNQ3 overlap, white and green arrows indicate AnkG-only labeling of proximal AIS. Scales: 50 μm, upper; 10 μm, middle and lower.

Source data 2. Single marker grey-scale images and selected merged images of CA1. In middle and lower panels, the yellow boxed in upper panels is shown as single channel greyscale and as the indicated merges of 2 channels. In lower merged images, yellow arrows indicate portions of AISs showing KCNQ2/KCNQ3 overlap, and white and green arrows indicate AnkG-only labeling of proximal AISs. Scales: 50 μm, upper; 10 μm, middle and lower.

Source data 3. Single marker grey-scale images and selected merged images of CA3. Scales: 50 μm.

MDAR checklist

### Data availability

All data generated or analysed during this study are included in the manuscript and supporting files.

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

# Appendix 1

## Appendix 1—key resources table

| Reagent type (species) or resource | Designation | Source or reference | Identifiers | Additional information |
|---|---|---|---|---|
| Antibody | Rabbit anti-KCNQ2N1 | PMID: 11739564, PMID: 14762142 | RRID:AB_2314688 | IHC (1:200) WB (1:400) |
| Antibody | Guinea pig anti-KCNQ3N1 | PMID: 16525039, PMID: 18827480 | RRID:AB_3303619 | IHC (1:500) WB (1:1000) |
| Antibody | Mouse anti-PanNav igG1 | Sigma Aldrich | Catalog #: S8809 RRID:AB_477552 | IHC (1:200) |
| Antibody | Anti-α-Tubulin antibody, Mouse monoclonal | Sigma Aldrich | Catalog #: T6199 RRID:AB_477583 | WB (1:10,000) |
| Antibody | Mouse anti-Ankyrin G IgG2a | Neuromabs | Clone N106/36.1 RRID:AB_10697718 | IHC (1:1000) |
| Antibody | Goat anti-Rabbit IgG Alexa Fluor Plus 555 | Invitrogen | Catalog #: A32732 UH287772 RRID:AB_2633281 | IHC (1:500) |
| Antibody | Cy5 AffiniPure Donkey Anti-Guinea Pig IgG (H+L) | Jackson ImmunoResearch | Code: 706-175-148 Lot #: 144177 RRID:AB_2340462 | IHC (1:500) |
| Antibody | Goat anti-Mouse IgG1 Cross-Adsorbed Secondary Antibody, Alexa Fluor 488 | Invitrogen | Catalog #: A-21121 Lot #: 2083196 RRID:AB_2535764 | IHC (1:500) |
| Antibody | Goat anti-Mouse IgG2a Cross-Adsorbed Secondary Antibody, Alexa Fluor 488 | Invitrogen | Catalog #: A-21131 Lot #: 1145167 RRID:AB_2535771 | IHC (1:500) |
| Antibody | Peroxidase IgG Fraction Monoclonal Mouse Anti-Rabbit IgG, light chain specific | Jackson ImmunoResearch | Code: 211-032-171 Lot #: 97224 RRID:AB_2339149 | WB (1:5000) |
| Antibody | Peroxidase AffiniPure F(ab')$_2$ Fragment Goat Anti-Guinea Pig IgG (H+L) | Jackson ImmunoResearch | Code: 106-036-003 Lot #: 121003 RRID:AB_2337405 | WB (1:5000) |
| Antibody | Peroxidase AffiniPure Goat Anti-Mouse IgG, light chain specific | Jackson ImmunoResearch | Code: 115-035-174 Lot #: 12150 RRID:AB_2338512 | WB (1:10,000) |
| Cell line | Flp-In-CHO (Chinese hamster ovary) cell line | Invitrogen/Thermo Fisher Scientific | Catalog #: R75807 Lot #: 1819218 (Oct 2017) | Used in automated patch experiments |
| Cell line | CHO (Chinese hamster ovary) cell line | PMID: 15862463 | | Used in manual patch expts |
| Transfected construct | Human KCNQ2 cDNA in pcDNA3 | PMID: 9872318; PMID: 16525039 | NP_742106 | Manual patch expts; gift of Thomas Jentsch |
| Transfected construct | Human KCNQ3 cDNA in pcDNA3 | PMID: 9872318; PMID: 16525039 | NM_004519.4 | Manual patch expts; gift of Thomas Jentsch |
| Transfected construct | Human KCNQ2 cDNA in pIRES2-EGFP | PMID: 35104249 | NM_172108 | Automated patch expts |
| Transfected construct | Human KCNQ3 cDNA in pcDNA5/FRT | PMID: 35104249 | NM_004519.4 | Automated patch expts; from Thomas Jentsch |
| Strain, strain background (*Mus musculus*) | Kcnq2$^{G256W}$ (C57BL/6J-Kcnq2$^{em4(G256W)Lutzy}$/J) | This paper; Jackson Laboratory Mutant Mouse Resource and Research Center | JR 29407 | Heterozygous G256W mice |

*Appendix 1 Continued on next page*

*Appendix 1 Continued*

| Reagent type (species) or resource | Designation | Source or reference | Identifiers | Additional information |
|---|---|---|---|---|
| Strain, strain background (*Mus musculus*) | Kcnq2$^{E254fs*16}$ (C57BL/6J-Kcnq2$^{em5(7del)}$ $^{Lutzy}$/J) | This paper; Jackson Laboratory Mutant Mouse Resource and Research Center | JR 29408 | Heterozygous E254fs mice |
| Recombinant DNA reagent | pIRES2-EGFP mammalian expression vector | BD Biosciences-Clontech | | |
| Recombinant DNA reagent | pcDNA3 mammalian expression vector | ThermoFisher Scientific | | |
| Recombinant DNA reagent | pcDNA5/FRT mammalian expression vector | ThermoFisher Scientific | | |
| Chemical compound, drug | ProLong Gold Antifade Mountant | ThermoFisher Scientific | Cat. No. P36931 | |
| Commercial assay, kit | RNeasy Lipid Tissue Mini Kit | Qiagen | Cat. No. 74804 | |
| Commercial assay, kit | SuperScript III First-Strand Synthesis SuperMix | Invitrogen | Ref 18080–400 | |
| Commercial assay, kit | TaqMan Fast Advanced Master Mix for qPCR | Applied Biosystems | Cat. No. 4444557 | |
| Commercial assay, kit | Kcnq2 Taqman gene expression assay | Thermo Fisher Scientific | Assay ID: Mm00440084_mH | Allele non-specific assay |
| Commercial assay, kit | Kcnq2 Taqman gene expression assay | Thermo Fisher Scientific | Assay ID: AP2XHY6 | WT specific assay |
| Commercial assay, kit | Kcnq3 Taqman gene expression assay | Thermo Fisher Scientific | Assay ID: Mm00548884_m1 | |
| Commercial assay, kit | Gapdh Taqman gene expression assay | Thermo Fisher Scientific | Assay ID: Mm99999915_g1 | |
| Commercial assay, kit | QuantStudio 3 Real-Time PCR System, 96-well, 0.2 mL, laptop | Thermo Fisher Scientific | A28567 | |
| Sequence-based reagent | Kcnq2 exon 4–7 region forward primer | This paper | PCR Primers | 5'-CGG TAG TCT ACG CTC ACA GC-3' |
| Sequence-based reagent | Kcnq2 exon 4–7 region reverse primer | This paper | PCR Primers | 5'-TCT TGG ACT TTC AGG GCA AA–3' |
| Sequence-based reagent | Kcnq2 G256W mutagenic forward primer | This paper | PCR Primers | 5'-gcagagaaaTggg agaacgaccacttt gacacctac–3' |
| Sequence-based reagent | Kcnq2 e G256W mutagenic reverse primer | This paper | PCR Primers | 5'-ttctcccAtttctctgcc aagtacacc aggaacgag-3' |
| Commercial assay, kit | DNA Clean & Concentrator –25 Kit | Zymo Research | D4033 | |
| Commercial assay, kit | ApexHot Start 2 X Master Mix Blue Apex Buffer 1 | Genesee Scientific | 42–143 | |
| Chemical compound, drug | 4 X Protein Sample Loading Buffer for Western Blots | Li-Cor | Selected P/N: 928–40004 | |
| Chemical compound, drug | Precision Plus Protein Standards All Blue | Biorad | 161–0373 | |
| Commercial assay, kit | Chameleon Duo Pre-stained Protein Ladder | Li-Cor | 928–60000 | |
| Commercial assay, kit | 7.5% Mini-PROTEAN TGX Precast Protein Gels, 12-well, 20 µl | Biorad | #4561025 | |
| Chemical compound, drug | Pierce protease inhibitor mini tablets, EDTA-free | Thermo Fisher Scientific | Ref: A32955 | |

*Appendix 1 Continued on next page*

*Appendix 1 Continued*

| Reagent type (species) or resource | Designation | Source or reference | Identifiers | Additional information |
|---|---|---|---|---|
| Commercial assay, kit | ECLPrime Western Blotting Detection Reagents | Cytiva Life Sciences | RPN2232 | |
| Commercial assay, kit | Amersham Hybond 0.2 μm PVDF | Amersham | Cat. No. 10600021 | |
| Commercial assay, kit | BCA Protein Assay Kit | Pierce | Prod # 23227 | |
| Commercial assay, kit | CoverWell Incubation Chamber Gasket | Thermo Fisher Scientific | Cat. No. C18150 | |
| Commercial assay, kit | Digital Sonifier 450 | Branson | Cat. No. 15338553 | |
| Commercial assay, kit | 1/8" Sonifier Doublestep Microtip | Branson | Cat. No. 101063212 | |
| Commercial assay, kit | MaxCyte STx electroporator | MaxCyte | | |
| Commercial assay, kit | Axopatch 200B amplifier | Molecular Devices | | |
| Commercial assay, kit | Multiclamp 700B amplifier | Molecular Devices | | |
| Commercial assay, kit | cFlow perfusion controller | Cell MicroControls | | |
| Commercial assay, kit | mPre8 manifold | Cell MicroControls | | |
| Commercial assay, kit | Leica VT1200S | Leica | | |
| Commercial assay, kit | Single Channel Temperature Controller | Warner Instruments | Cat. No. TC-324C | |
| Software, algorithm | Prism 9.0 | GraphPad | RRID:SCR_002798 | |
| Software, algorithm | NIS-Elements | Nikon | RRID:SCR_002776 | |
| Software, algorithm | Snapgene | Snapgene | RRID:SCR_015052 | |
| Software, algorithm | Image Studio v5.2 | Li-Cor | RRID:SCR_015795 | |
| Software, algorithm | Pclamp/ Clampfit | Molecular Devices | RRID:SCR_011323 | |
| Software, algorithm | PatchController384 V.1.3.0 | Nanion Technologies | | |
| Software, algorithm | Rstudio | Rstudio | RRID:SCR_000432 | |
| Software, algorithm | Chimera-1.16-win64; ChimeraX-1.6.1 | UCSF | RRID:SCR_004097; RRID:SCR_015872 | |
| Software, algorithm | LabChart | AD Instruments | RRID:SCR_017551 | EEG data analysis |

