## [Editor Report · eLife assessment]

The paper investigates a potential cause of a type of severe epilepsy that develops in early life because of a defect in a gene called KCNQ2. The significance is **fundamental** because it substantially advances our understanding of a major research question. The strength of the evidence is **convincing** because appropriate methods are used that are in line with the state-of-the art.

---

## [Referee Report · Reviewer #1 (Public Review)]

Abreo et al., performed a detailed multidisciplinary analysis of a pathogenic variant of the KCNQ2 ion channel subunit identified in a child with neonatal-onset epilepsy and neurodevelopmental disorders. These analyses revealed multiple molecular and cellular mechanisms associated with this variant, and providing important insights into what distinguishes distinct pathogenic variants of KCNQ2 associated with self-limited familial neonatal epilepsy versus those leading to developmental and epileptic encephalopathy, and how they may mechanistically differ, to result in different extents of developmental impairment. The authors first provide a detailed clinical description of the patient heterozygous for a novel pathogenic variant encoding KCNQ2 G256W. They then model the structure of the G256W variant based on recent cryo-EM structures of KCNQ2 and other ion channel subunits and find that while the affected position is quite distinct from the channel pore, it participates in a novel, evolutionarily conserved set of amino acids that form a network of hydrogen bonds that stabilize the structure of the pore domain. They then undertake a series of rigorous and quantitative laboratory experiments in which the KCNQ2 G256W variant is coexpressed exogenously with WT KCNQ2 and KCNQ3 subunits in heterologous cells, and endogenously in novel gene edited mice generated for this study. This includes detailed electrophysiological analyses in the transfected heterologous cells revealing the dominant-negative phenotype of KCNQ2 G256W. They find altered firing properties in hippocampal CA1 neurons in brain slices from the heterozygous KCNQ2 G256W mice. They next show that the expression and localization of KCNQ channels is altered in brain neurons from heterozygous KCNQ2 G256W mice, suggesting that this variant impacts KCNQ2 trafficking and stability. Together, these laboratory studies reveal that the molecular and cellular mechanisms shaping KCNQ channel expression, localization and function are impacted at multiple levels by the variant encoding KCNQ2 G256W, likely contributing to the clinical features of the child heterozygous for this variant relative to patients harboring distinct KCNQ2 pathogenic variants.

---

## [Referee Report · Reviewer #3 (Public Review)]

Summary:

This manuscript describes the symptoms of patients harboring KCNQ2 mutation G256W, functional changes of the mutant channel in exogenous expression, and phenotypes of G256W/+ mice. The patients presented seizures, the mutation reduced currents of the channel, and the G256W/+ mice show seizures, increased firing frequency in neurons, and reduced KCNQ2 expression and altered subcellular distribution.

Strengths:

This is a large amount of work and all results corroborated the pathogenicity of the mutation in KCNQ2, providing an interesting example of KCNQ2-associated neurological disorder's impact on functions at all levels including molecular, cellular, tissue, animal model and patients.

---

## [Author Response]

The following is the authors’ response to the original reviews.

**eLife assessment**
The paper investigates a potential cause of a type of severe epilepsy that develops in early life because of a defect in a gene called KCNQ2. The significance is fundamental because it substantially advances our understanding of a major research question. The strength of the evidence is convincing because appropriate methods are used that are in line with the state-of-the art, although there are some revisions/corrections that would strengthen the evidence further.

Thank you for the expert, thorough, and helpful review. We believe that addressing the reviewers’ points has improved our paper greatly.

**Public Reviews:**

**Reviewer #1 (Public Review):**
Abreo et al. performed a detailed multidisciplinary analysis of a pathogenic variant of the KCNQ2 ion channel subunit identified in a child with neonatal-onset epilepsy and neurodevelopmental disorders. These analyses revealed multiple molecular and cellular mechanisms associated with this variant and provided important insights into what distinguishes distinct pathogenic variants of KCNQ2 associated with self-limited familial neonatal epilepsy versus those leading to developmental and epileptic encephalopathy, and how they may mechanistically differ, to result in different extents of developmental impairment.The authors first provide a detailed clinical description of the patient heterozygous for a novel pathogenic variant encoding KCNQ2 G256W. They then model the structure of the G256W variant based on recent cryo-EM structures of KCNQ2 and other ion channel subunits and find that while the affected position is quite distinct from the channel pore, it participates in a novel, evolutionarily conserved set of amino acids that form a network of hydrogen bonds that stabilize the structure of the pore domain.They then undertake a series of rigorous and quantitative laboratory experiments in which the KCNQ2 G256W variant is coexpressed exogenously with WT KCNQ2 and KCNQ3 subunits in heterologous cells, and endogenously in novel gene-edited mice generated for this study. This includes detailed electrophysiological analyses in the transfected heterologous cells revealing the dominant-negative phenotype of KCNQ2 G256W. They found altered firing properties in hippocampal CA1 neurons in brain slices from the heterozygous KCNQ2 G256W mice.They next showed that the expression and localization of KCNQ channels are altered in brain neurons from heterozygous KCNQ2 G256W mice, suggesting that this variant impacts KCNQ2 trafficking and stability.Together, these laboratory studies reveal that the molecular and cellular mechanisms shaping KCNQ channel expression, localization, and function are impacted at multiple levels by the variant encoding KCNQ2 G256W, likely contributing to the clinical features of the child heterozygous for this variant relative to patients harboring distinct KCNQ2 pathogenic variants.

Thank you for the thorough summary and estimation of the initial submission, we are very glad that our approach, analytical methods, and conclusions were convincing.

**Reviewer #2 (Public Review):**
Summary:The paper entitled "Plural molecular and cellular mechanisms of pore domain KCNQ2 encephalopathy" by Abreo et al. is a complex and integrated paper that is well-written with a focus on a single gene variant that causes a severe developmentalencephalopathy. The paper collates clinical outcomes from 4 individuals and investigates a variant causing KCNQ2-DEE using a wide range of experimental techniques including structural biology, in vitro electrophysiology, generation of genetically modified animal models, immunofluorescence, and brain slice recordings. The overall results provide a plausible explanation of the pathophysiology of the G265W variant and provide important findings to the KCNQ2-DEE field as well as beginning to separate the understanding between seizures and encephalopathies.Strengths:(1) The authors describe in detail how the structural biology of the channel with a mutation changes the movement of the protein and adds insights into how one variant can change the function of the M-current. The proposed model linking this change to pathogenic consequences should help pave the way for additional studies to further support this type of approach.(2) The multiple co-expression ratio experiments drill down to the complex nature of the assembly of channels in over-expression systems and help to move toward an understanding of heterozygosity. It might have been interesting if TEA was tested as a blocker to better understand the assembly of the transfected subunits or possibly use vectors to force desired configurations.(3) The immunofluorescent approach to understanding re-distribution is another component of understanding the function of this critical current. The demonstration that Q2 and Q3 are diminished at the AIS is an important finding and a strength to the totality of the data presented in the paper.(4) Brain slice work is an important component of studying genetically modified animals as it brings in the systems approach, and helps to explain seizure generation and EEG recordings. The finding that G265W/+ neurons were more sensitive to current injections is a critical component of the paper.(5) The strength of this body of work is how the authors integrated different scientific approaches to knitting together a compelling set of experiments to better explain how a single variant, and likely extrapolation to other variants, can cause a severe neonatal developmental encephalopathy with a poor clinical outcome.

Thank you for the thorough and encouraging reading of our work and its strengths, we are very glad that, excepting the issues mentioned which we have addressed, our approach and conclusions were convincing.

Weaknesses:(1) Minor comment: Under the clinical history it is unclear whether the mother was on Leviracetam for suspected in-utero seizures or if Leviracetam was given to individual 1.The latter seems more likely, and if so this should be reworded.

We revised the results text to clarify that the drug was begun postnatally, after epilepsy was diagnosed in the child.

(2) As described in the clinical history of patient 1, treatment with ezogabine was encouraging with rapid onset by a parental global impression with difficulty in weaning off the drug. When studying the genetically modified mice, it would have been beneficial to the paper to talk about any ezogabine effects on the genetically modified mice.

We agree this is of great interest, but sampling and metrics are challenging due to the very low frequency of seizures and delayed mortality in the heterozygous G256 mice. Accordingly, we have not performed ezogabine treatment experiments in the mice described in this study, which model a human variant associated with a brief neonatal window of frequent seizures. We hope to return this issue using other transgenic mice with higher seizure frequency, but such results are outside the current scope.

(3) It is a bit surprising that CA1 pyramidal neurons from the heterozygous G256W mice have no difference in resting membrane potential. The discussion section might explore this in a bit more detail.

Thank you for raising this issue. This combination of outcomes has been seen previously and is interpreted as an outcome of low somatodendritic surface expression of the channels. Relatively higher expression within the AIS membrane, with its the relatively small surface area and electrical isolation from the soma, allow the KCNQ2/3 channels to influence AIS excitability with little or (in this instance) undetectable influence on the RMP (see e.g., Otto et al. 2006, PMID: 16481438; Singh et al. 2008, PMID 16481438 for KCNQ2 mutant mice. See Hu and Bean, 2018, figure 2; PMID: 29526554 for explicit testing via focal AIS vs. somatic blocker perfusion). Additionally, in previous work, we did not find any changes to the RMP of CA1 pyramidal neurons in either Kcnq2 knockout mice (PMID: 24719109) or mice expressing a Kcnq2 GOF variant (PMID: 37607817). We modified the discussion including adding references to prior studies combining experimental and multicompartmental computational models.

(4) It was mentioned in the paper about a direct comparison between SLFNE and G256W.However, in the slice recordings, there was no comparison. Having these data comparingSLFNE to G256W would have been a more fulsome story and would have added to the concept around susceptibility to action potential firing.

Thank you for this point. We agree that such side-by-side recordings would be interesting. However, slice recordings were not performed on the SLFNE mice. The study design was based on the fact that extensive prior studies of both haploinsufficient and missense human SLFNE variant mice have been published (Otto et al. 2006 J Neuroscience, PMID: 16481438; Singh et al. 2008, PMID 16481438; Kim et al 2020 PMID: 31283873) and show good agreement, but DEE missense variants have not been previously studied. We revised the discussion, to place the current DEE model results in the context of the prior SNFLE model slice work. We contrast the similarity of the CA1 cellular hyperexcitability phenotype ex vivo (at least in CA1 pyramidal cells) across models to the differences in electrographic and behavioral seizures (i.e., network level physiology).

**Reviewer #3 (Public Review):**
Summary:This manuscript describes the symptoms of patients harboring KCNQ2 mutation G256W, functional changes of the mutant channel in exogenous expression, and phenotypes of G256W/+ mice. The patients presented seizures, the mutation reduced currents of the channel, and the G256W/+ mice showed seizures, increased firing frequency in neurons, reduced KCNQ2 expression, and altered subcellular distribution.Strengths:This is a large amount of work and all results corroborated the pathogenicity of the mutation in KCNQ2, providing an interesting example of KCNQ2-associated neurological disorder's impact on functions at all levels including molecular, cellular, tissue, animal model, and patients.Weaknesses:The manuscript described observations of changes in association with the mutation at molecular cellular functions and animal phenotype, but the results in some aspects are not as strong as in others. Nevertheless, the manuscript made overarching conclusions even when the evidence was not sufficiently strong.

Thank you for your review. In our revision (as listed in the recommendations to authors section) we have attempted to better justify the conclusions you mention there.

**Recommendations for the authors:**

**Reviewer #1 (Recommendations For The Authors):**
Suggestions for improved or additional experiments, data, or analyses.Page 7: the authors' statement that G256 could be intolerant to substitution would be strengthened by a straightforward analysis of available genome- and exome-wide sequencing data to determine the level of genic intolerance at this position in the human population, as has been used previously to highlight critical residues including those impacted by pathogenic variants in many other proteins including ion channels (e.g., Genome Biology 17:9, 2016; Am J Hum Genet 99:1261, 2016; Biochim Biophys Acta Biomemb 1862:183058, 2020).

Thank you for this suggestion, we have revised the opening of this section to point out the low ratio of benign to pathogenic variants in the region surrounding G256 shown by prior work. We have added citations to the papers describing the MTR and gnomAD tools that highlight these data and calculations.

The overall interpretation of the CHO cell results would be enhanced by the authors including in their discussion an explicit statement that they did not attempt to evaluate the overall and plasma membrane expression levels of the exogenously expressed WT and mutant KCNQ2 subunits, nor that of KCNQ3, in the transfected CHO cells. They could also highlight that this is an important future experiment to determine whether the dominant negative effects are due to impaired expression/trafficking or impaired function of plasma membrane channels, as this may be an important consideration for designing therapeutic strategies.

We agree. We revised the discussion to explicitly mention this additional direction. We agree this topic has therapeutic implications, especially given our in vivo protein localization results. We added a mention that combinations of molecules enhancing surface localization with channel openers could be a therapeutic strategy, analogous to approved therapies for cystic fibrosis.

The authors conclude that the impact of ezogabine treatment is reduced in the cells expressing G256+/W versus those expressing WT KCNQ2. However, the delta pA/pF graph in panel 3G expresses the effects of ezogabine as absolute increases in current density. Determining the relative increase (i.e., fold change) in current density in ezogabine-treated versus control conditions is a more valid way to analyze these data. This provides a better reflection of the impact of ezogabine as the control currents already have a much larger amplitude than the G256+/W currents. By eye the impact of ezogabine looks comparable or even larger for the G256+/W condition than for WT, fundamentally changing the interpretation of these results.

Thank you for this helpful comment. The reviewer calls attention to the fact that although G256W/+ mean whole cell currents from are less than WT, before and after application of ezogabine, it appeared from Fig. 3G that ezogabine enhanced currents to a “proportionally equivalent extent” in G256W/+ and WT cells. We revised panel 3G to try to make this more clear. It now shows WT currents +/- ezogabine currents normalized to (WT, no ezogabine at +40 mV), along with G256W/+ cells +/- ezogabine currents, normalized to (G256W/+, no ezogabine at +40 mV). This normalization shows that the mixed population of channels expressed by G256W/+ cells are equally augmented (with a trend toward greater augmentation), compared to controls. This is a striking result given that channels lacking WT KCNQ2 subunits do not respond to ezogabine (i.e., the “homozygous heteromer” condition, Fig. 3F) do not respond to ezogabine. Although the underlying data are unchanged, we agree with the reviewers’ conclusion about emphasizing the effect “per channel”. This reframing is mechanistically and clinically important. We have made changes to the results text and discussion to highlight related issues.

Figure 7: it is not clear from the information presented whether the qPCR would only measure WT KCNQ2 mRNA levels or detect levels of both WT and E254fs transcripts. The authors assume nonsense-mediated decay, but they did [not] determine experimentally that this occurred. The sequencing in the supplemental figure shows the presence of E254fs transcripts but does not allow for insights into their abundance. It should be straightforward to develop primer sets that could then be used to selectively amplify WT and E254fs transcripts for quantitation.

Thank you for this helpful suggestion. The assay used in the initial submission measures total Kcnq2 mRNA. We developed and performed a new assay where the probe binding site is the WT sequence, centered on the mutations. New Figure 7-Figure supplement 1, panel A is a cartoon showing the differences between the assays. Using the WT alleleselective RT-qPCR assay, both G256W/+ and E254fs/+ samples showed a 50% loss of WT Kcnq2. We now can conclude that NMD is absent for G256W and incomplete for E254fs mRNA. Neither mutant heterozygous line shows a compensatory increase in WT Kcnq2 expression. These conclusions are much more specific than previously, and documenting incomplete NMD of KCNQ2 is novel and of potential clinical significance. The KCNQ2 protein (western blot) and WT mRNA (qPCR) results now agree, both showing ~50% loss.

For reporting transparency, the authors should provide the sequences of each of the primers used. Perhaps this is in the "key reagents" section, but this was missing from the manuscript. I note the authors use NMD in this section without defining it. and added a reference to a review where “incomplete NMD” is discussed.

We have added the assay catalogue numbers to the key reagents table. We eliminated the use of the NMD abbreviation. We added citations to the “incomplete NMD” literature including an excellent recent review and a directly relevant primary paper. These show how NMD efficiency may differ: between genes, transcripts, cells, tissues and, remarkably, between human individuals (see doi: 10.1093/hmg/ddz028, cited in the review—caffeine inhibits NMD!). The revised discussion mentions this, and relevance to future studies of novel KCNQ2 variant pathogenicity and severity prediction.

Recommendations for improving the writing and presentation.I found the presentation of the IHC images deficient in terms of accessibility and transparency. While the movies provided are also useful, it is important the authors also provide conventional static merged images of each of their multiplex labeling images in the body of the paper. This allows a reader to see the labeling with the different antibodies in the context of each other (one of the major advantages of multiplex labeling), instead of trying to remember the pattern each label gave in prior sections of the movie.[We queried the reviewer via the eLife editorial staff]: To clarify my suggestion to improve Figure 8, the authors should generate from their movies static images that are basically what they already did in Fig8S3 for the G256W Het panel of the Fig8 movie. This involves revising Fig8S3 to include WT panels, and adding two new supplemental figures that show WT/Het panels with the separate antibodies and then a merged image from Fig8S1 and Fig8S2, just like they did in Fig8S3 for the mutant part of the Fig8 movie.

Thank you for this comment. As suggested by the reviewer, for each IHC movie (Fig. 8, Fig. 8-figure supplement 1 and Fig. 8-figure supplement 2), we added a new supplementary figure showing WT and mutant animal static images corresponding to the movies. For main Figure 8 (CA1, G256W/+ comparison), the new static images enable evaluating the patterns of colocalization by providing selected portions of the images at the highest useful magnification. These show each individual antibody in greyscale (best for comparing) and 4 different green-red merged images to show overlap (yellow) vs non-overlap. The merged images demonstrate colocalization of KCNQ2 and KCNQ3 at the distal portions of AnkG-labelled CA1 pyramidal cell AISs, in agreement with prior publications. In G256W/+ but not E254fs/+ images, KCNQ2 and KCNQ3 show reduced relative labeling of AISs and increased relative labeling of somata in the pyramidal cell layer. For CA3, the merged views show the redistributed relative labeling of KCNQ2 and KCNQ3 between stratum lucidum and stratum pyramidale.

We also revised Fig. 8 supplement 3 (CA1) to include WT panels, On reexamination, all WT interneurons in the small sample lacked somatic KCNQ2 and KCNQ3 labeling. Some s. oriens and radiatum AISs of both WT and G256W/+ sections showed KCNQ2 and KCNQ3 labeling, as shown in the revised figure. Counting statistics are included in the supporting data. Importantly, our belief that the images shown are representative is supported by the blinded analysis of a much larger sample (Figure 9, unchanged in revision).

Dragging the movie viewer “slider” allows the viewer to move back and forth between color channels. It works well in eLife if used in that way. This is a way of seeing the “representativeness” of the merges shown in the CA1 conventional static images, which necessarily include a smaller x-y area and include only a few AISs. We also added a KCNQ2/KCNQ3 merge to the movies.

Western blot results in Figure 9 - Supplement 1: for transparency, the authors need to show the entire blot, as they did in Figure 4 - Supplement 2. This is required in many journals, and in the case of KCNQ2 it provides crucial information as to the different forms of KCNQ2 present on SDS gels in these samples that contain different KCNQ2 isoforms. Given the surprising decrease in levels of KCNQ2 monomer in the G256+/W mice, it is important to present and analyze the levels of the monomer, dimer, and higher oligomeric forms of KCNQ in these samples, to determine whether protein "missing" in the monomeric form is not present in the dimeric or higher oligomeric form. This is especially important as the G256W mutant could lead to misfolding and aggregation leading to a higher proportion of both WT and G256W subunits being present in a higher-order oligomeric form. I note that it is odd that the figure legend states "Images of entire filter used for western blot of lysates, probed for KCNQ2 and KCNQ3.", even though only selected portions are shown.

Thank you for this suggestion. We agree that the wording of the legend needed improvement.

In revision, the western blots are renumbered as Figure 10, and Figure 10-Figure supplement 1. In the main figure, monomer bands and densitometry are shown, as previously. In the new Figure 10-Figure supplement 1, we show (1) the ECL image of the entire filter probed with rabbit anti-KCNQ2, (2) the same blot, stripped, and reprobed with guinea pig KCNQ3, (3) the lower portion, probed with mouse anti-tubulin. The revised Fig. 10-fig supplement 1 shows 3 genotypes x 3 individual (male) p21 mice, with all steps performed in parallel from homogenization to ECL detection. As suggested, we performed new analysis of the immunoreactive bands corresponding to (apparent) monomer, dimer, and higher oligomeric forms of KCNQ2. Analysis of the sum of those bands showed loss of KCNQ2 protein in both mutant lines.

The methods are sufficiently detailed with the exception that there is inconsistent inclusion of catalog numbers and RRIDs. Having these would improve transparency as to specific reagents used and would allow for enhanced reproducibility of the lab research performed here.

The revised submission includes the key resources table, which we understood was not requested from eLife at initial submission.

Minor corrections to the text and figures.Typos/mistakes as to antibodies used in the IHC methods section "anti-AnkG36 N106/36 " should be "anti-AnkG N106/36", and "mouse anti-PanNav IgG1 supernatant" should be mouse anti-PanNav IgG1 purified antibody".

Thank you, corrections made.

It would facilitate a reader's interpretation of the IHC results if the authors explicitly stated in the IHC results section that the KCNQ2 antibody used is against the N-terminus and therefore should recognize both mutant isoforms as the mutations are downstream of this.

We added this point to the results section in relation to Figure 4-figure supplement 2 (western), and in IHC methods.

PV is not defined when used in the discussion, nor is why knowing that somatic KCNQ2 immunolabeling is present in both PV and non- PV interneurons of WT mice of value to the reader.

We revised these sentences for clarity.

The IHC methods state that "mice were transcardially perfused with....ice cold 2% paraformaldehyde in PBS, freshly prepared from a 20% stock (Electron Microscopy Sciences).". The authors presumably mean "formaldehyde" as paraformaldehyde is the inert polymeric storage form of active depolymerized monomeric formaldehyde that is a fixative.

The reviewer is correct regarding the chemistry; the manufacturer’s product name is “Paraformaldehyde 20% aqueous solution”. We revised accordingly.

**Reviewer #3 (Recommendations For The Authors):**
Some comments regarding the presentation are as follows.(1) The section "G256W lies atop a dome-shaped hydrogen bond network linking helix S5 to the turret and selectivity filter" is entirely based on structural observations without functional validation. This may be more appropriate in Discussion. The emphasis on the "turret arch" bonding should be tuned down due to the lack of functional support.

We understand and agree with this concern about the distinction between structural analysis and implied function. However, we believe that the structural model reinterpretation and phylogenetic sequence analysis in our submission are results. Structures as complex as those of KCNQ channels necessarily cannot be fully shown or analyzed in an initial publication. To our knowledge, the word “turret” has not appeared in a KCNQ channel cryoEM paper to date. Bringing clinical motivation to prioritize study of an overlooked spot on the channel is creditworthy. The comprehensive heterologous patch clamp results in our study (including absence of effects on voltage-dependence, evidence of partial functional activity of channels containing one mutant subunit per channel shown for KCNQ2 homomers, KCNQ2/3 heteromers, and via acute ezogabine rescue experiments in the biologically most relevant heteromers) are functional evidence consistent with G256W acting through disruption of the SF.

However, we agree that more support is needed. The words “dome” and “arch”, though accurate for describing shape, tend to imply a mechanical “load bearing and distributing” function --our study does not prove this. Accordingly, we have toned down the emphasis by removing the words “keystone”, “turret dome bonding”, and “as a structural novelty” from the abstract. The revised discussion section replaces arch with “arch-shaped”, calls the idea that the turret functions as a stabilizing arch a “novel hypothesis”, and proposes next experiments (with relevant citations).

Section title "Heterozygous G256W mice have neonatal seizures" does not seem to match the results since there was only one mouse that showed neonatal seizures.

Thank you, we have revised the section title. The text is transparent regarding sample size. The discussion highlights that these seizures are rare (indeed, not previously shown for any heterozygous missense model, to our knowledge).

(2) It will be nice for the non-expert readers if the observations of "discrete seizures", "clusters", "diffuse bilateral onset", "unilateral onset" etc. are marked in Figure 1.

Thank you for making this point. Figure 1 shows key excerpts of one bilateral onset seizure; a unilateral onset example isn’t shown since previous KCNQ2 DEE papers we cite have emphasized and illustrated focal onset seizures (Weckhuysen et al., 2013; Numis et al., 2014). We revised the results section (p. 4) and Figure 1 and supplement captions to improve clarity for all readers including non-specialists.

(3) Figure 5 and page 10 first paragraph. Please specify the number of cells and the number of mice that were studied.

Thank you, this information has been added to legend.